# UniZyme: A Unified Protein Cleavage Site Predictor Enhanced with Enzyme Active-Site Knowledge

**Chenao Li, Shuo Yan, Enyan Dai**[*]
Hong Kong University of Science and Technology (Guangzhou)

## Abstract

Enzyme-catalyzed protein cleavage is essential for many biological functions. Accurate prediction of cleavage sites can facilitate various applications such as drug development, enzyme design, and a deeper understanding of biological mechanisms. However, most existing models are restricted to an individual enzyme, which neglects shared knowledge of enzymes and fails to generalize to novel enzymes. Thus, we introduce a unified protein cleavage site predictor named UniZyme, which can generalize across diverse enzymes. To enhance the enzyme encoding for the protein cleavage site prediction, UniZyme employs a novel biochemically-informed model architecture along with active-site knowledge of proteolytic enzymes. Extensive experiments demonstrate that UniZyme achieves high accuracy in predicting cleavage sites across a range of proteolytic enzymes, including unseen enzymes. The code is available in `https://github.com/Ao-LiChen/UniZyme`.

## 1 Introduction

During enzyme-catalyzed protein hydrolysis, proteolytic enzymes cleave proteins at specific cleavage sites. This process is illustrated in Fig. 1, and it is crucial for a variety of physiological processes, including cell proliferation, immune response, and cell death [1, 2]. Accurate prediction of enzyme-catalyzed cleavage sites in the substrate proteins facilitates the identification of therapeutic targets and guides drug design [3]. For instance, abnormal protein hydrolysis is closely associated with cancer, viral infections, and neurodegenerative diseases, and predicting the cleavage sites of abnormal proteins under pathological conditions can reveal biomarkers or intervention targets [4, 5]. Additionally, in the design of enzyme inhibitors or prodrugs, identifying key cleavage peptides, such as those cleaved under the catalysis of HIV enzyme, helps enhance drug specificity and minimize off-target effects [6, 7].

Mapping cleavage sites experimentally via peptide assays or high-throughput mass spectrometry is arduous and costly [8]. Therefore, recent studies have employed machine learning methods to advance the prediction of protein cleavage sites. For example, CAT3 [9] predicts caspase-3 cleavage sites based on position-specific scoring matrices (PSSM), and ProsperousPlus [10] integrates multiple methods to comprehensively evaluate cleavage site predictions. However, these methods generally focus on an individual enzyme system, overlooking shared patterns and failing to generalize to proteases without labeled data. This limitation impedes tasks like off-target

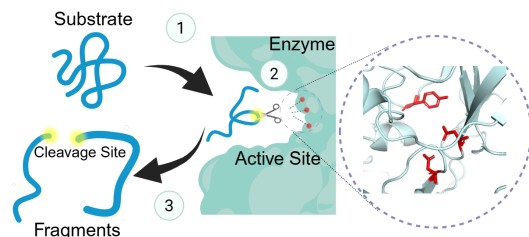

Figure 1: Enzyme-catalyzed protein hydrolysis.

---

[*]Corresponding author: `enyandai@hkust-gz.edu.cn`

39th Conference on Neural Information Processing Systems (NeurIPS 2025).

assessment of therapeutic proteins in human body [11, 12]. Therefore, it is crucial to develop a unified protein cleavage site predictor that can generalize across a diverse range of proteolytic enzymes.

To develop a unified protein cleavage site predictor for diverse proteolytic enzymes, the information of enzyme should be extracted and incorporated for the prediction. However, due to substantial cost of biological experiments, existing cleavage site databases only cover a small number of proteolytic enzymes (Tab. 1), which significantly challenges the learning of enzyme information encoder. Despite the limited coverage of enzymes in existing cleavage site datasets, many proteolytic enzymes are annotated with their active sites, which is the core functional region for catalyzing the protein hydrolysis. Specifically, the unique physicochemical environment of these active sites enables recognition of target substrates and lowers the activation energy required for cleaving specific peptide bonds. Therefore, we propose to leverage redundant knowledge of enzyme active sites to enhance the modeling of enzymes in enzyme-catalyzed protein cleavage sites.

However, it is non-trivial to achieve a unified cleavage site predictor enhanced with enzyme active-site knowledge. Two major challenges remain to be resolved. *First*, the cleavage process is influenced by various factors of enzymes such as 3D structures and environments of active sites. Hence, how to design the architecture of enzyme encoder to effectively capture useful information for enzyme-catalyzed cleavage site prediction? *Second*, the active-site regions of enzymes determine the specificity and efficiency of enzymatic hydrolysis. How can we leverage this rich information of enzyme active sites to improve cleavage site prediction? In an attempt to address the challenges, we propose a novel framework named UniZyme. More specifically, a biochemically-informed enzyme encoder is deployed along with the active site-aware pooling to produce high-quality enzyme representations. We further augment the enzyme encoder by pretraining on a supplemented enzyme set for active-site prediction. Furthermore, a joint training of enzyme active-site prediction and substrate cleavage site prediction is applied in UniZyme. In summary, our main contributions are as follows:

- We investigate a novel and crucial problem of building a unified protein cleavage site predictor that generalizes across diverse proteolytic enzymes;
- We propose a novel framework UniZyme that effectively integrates the enzyme active-site knowledge to enhance the cleavage site prediction in enzyme-protein interaction;
- Extensive experiments demonstrate the effectiveness of our UniZyme in predicting cleavage sites of substrate proteins for both seen and unseen proteolytic enzymes.

## 2 Problem Formulation

In this section, we first introduce the preliminaries of enzyme-catalyzed protein hydrolysis. Then, we present the problem definition of protein cleavage site prediction with enzyme active-site knowledge.

### 2.1 Preliminaries of Enzyme-Catalyzed Protein Hydrolysis

**Cleavage Sites in Enzyme-Catalyzed Protein Hydrolysis**. *Protein hydrolysis* is a biochemical process where proteins are broken down into smaller fragments such as amino acids and peptides under the catalysis of proteolytic enzymes. As illustrated in Fig. 1, during the protein hydrolysis, proteolytic enzymes will firstly recognize specific amino acid sequences or structural motifs within substrate proteins. Then, the enzymes catalyze the cleavage of peptide bonds at the *cleavage site*, leading to the formation of smaller peptide fragments or individual amino acids. The positions of cleavage sites are governed by various factors including substrate's amino acid composition, spatial conformation, and unique properties of the enzyme [13–15].

**Active Sites of Enzymes**. The active sites in enzyme provide an environment that lowers the activation energy required for peptide bond cleavage. As shown in Fig. 1, with the active sites, enzymes can recognize and bind to target substrates, enabling the cleavage of specific peptide bonds within substrate proteins [16]. Hence, active site information can benefit the modeling of enzyme-catalyzed protein hydrolysis.

**Current Framework of Cleavage Site Prediction**. Recent studies have employed machine learning models to predict cleavage sites [17–21]. However, these methods generally train an independent model for each enzyme, which only predicts the cleavage sites of substrate proteins under the catalysis

of one specific enzyme. Specifically, let $\mathcal{P}^s$ denote the substrate protein, this enzyme-specific cleavage site predictor aims to learn the $f : \mathcal{P}^s \to \mathbf{c}^{e,s}$, where $\mathbf{c}^{e,s} \in \{0,1\}^{|\mathcal{P}^s|}$ denotes the labels of cleavage site with the enzyme $\mathcal{P}^e$. However, the training of enzyme-specific model excludes the valuable interaction knowledge from other enzyme–protein systems. In addition, the enzyme-specific model cannot generalize to unseen enzymes, which limits its application on unseen enzymes and other enzymes with limited annotations. Therefore, it is crucial to develop a unified cleavage site predictor capable of identifying cleavage sites in substrate proteins across various enzymes.

**Limited Enzyme Coverage in Cleavage Site Database**. Tab. 1 presents statistics from the MEROPS, which is the most comprehensive cleavage site

Table 1: Statistics of Cleavage Sites Data in MEROPS.

| # Proteolytic Enzyme | # Substrate | Enzyme–Substrate Ratio |
|---|---|---|
| 866 | 10,146 | 1:11.7 |

database. Due to the high cost of experimental assays, MEROPS [22] only includes 866 commonly used proteolytic enzymes. This results in a striking enzyme–substrate ratio of 1:11.7. The limited enzyme coverage poses a significant challenge for developing a unified cleavage site predictor that generalizes across diverse enzyme–substrate systems. Despite the limited enzyme coverage in cleavage site databases, the lower cost of annotating enzyme active sites has enabled UniProt [23] to provide 10,749 high-quality proteolytic enzymes with labeled active sites across multiple organisms. The rich information of active sites can be helpful in enzyme modeling to facilitate the cleavage site prediction in protein hydrolysis.

## 2.2 Problem Definition

In protein hydrolysis, both enzyme $\mathcal{P}^e$ and substrate $\mathcal{P}^s$ are proteins composed of amino acid residues that fold into 3D structures. We denote a protein of length $N$ by $\mathcal{P} = (\mathbf{X}, \mathbf{R})$, where $\mathbf{X} \in \mathbb{R}^{N \times d}$ is the feature matrix of $N$ residues, $\mathbf{R} \in \mathbb{R}^{N \times 3}$ denotes the 3D positions of residues. We denote $\mathbf{c}^{e,s} \in \{0,1\}^{|\mathcal{P}_s|}$ as the cleavage site label for the substrate protein $\mathcal{P}^s$ under the catalysis of enzyme $\mathcal{P}^e$. The training data for cleavage site prediction can be represented as $\mathcal{D}_c = \{(\mathcal{P}_i^e, \mathcal{P}_i^s, \mathbf{c}_i^{e,s})\}_{i=1}^{|\mathcal{D}_c|}$. In this work, we propose to enhance the cleavage site prediction with the active site information of enzymes. Hence, we will also utilize a set of enzymes labeled with active sites $\mathbf{a} \in \{0,1\}^{|\mathcal{P}_e|}$, which is denoted as $\mathcal{D}_a = \{(\mathcal{P}_i^e, \mathbf{a}_i)\}_{i=1}^{|\mathcal{D}_a|}$.

During the test phase, we will predict the cleavage site for each pair of test enzymes and substrates $(\mathcal{P}_t^e, \mathcal{P}_t^s)$. The active sites of test enzymes will not be available for inference. And the test enzyme $\mathcal{P}_t^e$ can be either seen ,i.e, $\mathcal{P}_t^e \in \mathcal{D}_c$ or unseen, i.e, $\mathcal{P}_t^e \notin \mathcal{D}_c$, which correspond to supervised setting and zero-shot setting, respectively. For each test substrate protein $\mathcal{P}_t^s$, its cleavage site with the test enzyme $\mathcal{P}_t^e$ is not included in the training set $\mathcal{D}_c$. With the above notations and descriptions, the formal definition of building a unified cleavage site predictor can be given by:

**Problem 1.** Given the dataset $\mathcal{D}_c$ annotated for cleavage site prediction and the supplemented dataset $\mathcal{D}_a$ containing enzymes active sites, we aim to obtain a unified cleavage site predictor:

$$f : (\mathcal{P}^e, \mathcal{P}^s) \to \mathbf{c}^{e,s}, \tag{1}$$

which can accurately predict cleavage sites of test proteins $\mathcal{P}_t^e$ under the catalysis of test proteolytic enzymes $\mathcal{P}_t^s$. Note that test enzymes can be either seen or unseen during the training phase.

## 3 Methodology

In this section, we give the details of the proposed UniZyme. As the Fig. 2 shows, apart from the substrate encoder, UniZyme deploys an enzyme encoder to enable the generalization of cleavage site prediction across various enzymes. In addition, active site information in protein hydrolysis is incorporated into enzyme encoder training to enhance cleavage site prediction. Two main challenges remain to be addressed: (i) how to design the enzyme encoder to preserve critical information for cleavage site prediction? (ii) how to leverage the rich information of enzyme active sites to improve the cleavage site prediction? To tackle the above challenges, our UniZyme deploys a biochemically-informed enzyme encoder which augments the graph transformer with enzyme energy frustration. Furthermore, UniZyme employs active site-aware pooling to preserve the enzyme's information crucial for protein hydrolysis. To facilitate enzyme representation learning, UniZyme first pretrains the enzyme encoder using active site prediction with the supplemented enzyme set. Then, a joint

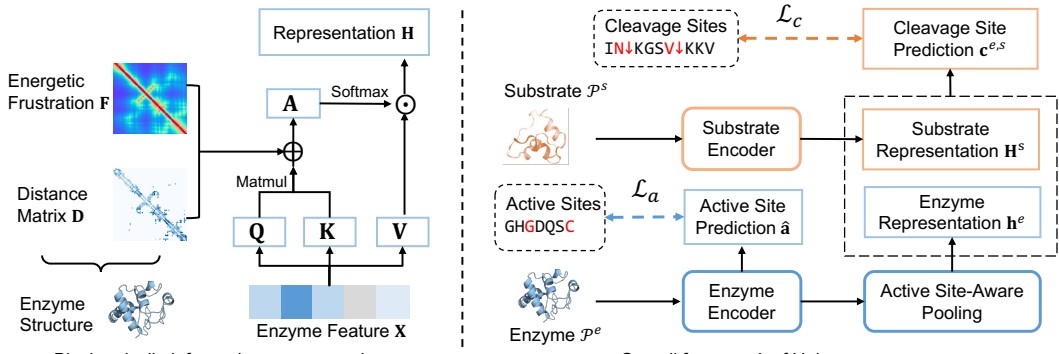

Figure 2: Architecture of biochemically-informed enzyme encoder and framework of UniZyme.

loss of active site prediction and cleavage site prediction is employed to optimize the UniZyme for accurate cleavage site prediction. Next, we introduce each component in detail.

## 3.1 Biochemically-Informed Enzyme Encoder

Enzymes' 3D structures, especially the local environments around their active sites, are crucial for catalyzing protein hydrolysis. Although direct active-site annotations are often unavailable for test data, recent studies indicate that local energetic frustration can identify functionally important regions [24]. Building on these insights, we propose a biochemically-informed enzyme encoder that integrates both the spatial positions of residues and their energetic frustration scores [25].

**Encoding Energetic Frustration.** Previous studies indicate that local energetic frustration, referring to regions in a protein not optimized for minimal energy, is commonly observed around enzyme active sites and can significantly influence catalysis [24]. To quantify this phenomenon, a frustration score $\mathbf{F}(i,j)$ is computed for each residue pair $(i,j)$ within an enzyme $\mathcal{P}^e$ following [24]:

$$\mathbf{F}(i,j) = \frac{\mathbf{E}(i,j) - \mu_{\text{rand}}(i,j)}{\sigma_{\text{rand}}(i,j)}, \tag{2}$$

where $\mathbf{E}(i,j)$ is the actual interaction energy between residues $(i,j)$ in the enzyme $\mathcal{P}^e$. $\mu_{\text{rand}}(i,j)$ and $\sigma_{\text{rand}}(i,j)$ represent the mean and standard deviation of interaction energies that derived from randomized configurations (see Appendix C for details). A higher $\mathbf{F}(i,j)$ implies stronger local energetic frustration, suggesting that the residue pair is more likely to belong to a functionally important region. Therefore, we incorporate this frustration score to provide useful biochemical information to the enzyme encoder.

**Integrating Energy and 3D Position in Transformer.** Following prior works [26], we encode the 3D positions by computing pair-wise distance between residues: $\mathbf{D}(i,j) = \|\mathbf{r}_i - \mathbf{r}_j\|_2$, where $\mathbf{r}_i \in \mathbb{R}^3$ is the C$\alpha$-atom coordinate of residue $i$. Both energetic frustration score and distance matrix capture pairwise relationships akin to the spatial encoding in graph transformers. Therefore, we locate those pair-wise signals to provide complementary information for the self-attention score computation. Concretely, for an enzyme $\mathcal{P}^e = (\mathbf{X}, \mathbf{R})$, we process both $\mathbf{F}(i,j)$ and distance matrix $\mathbf{D}(i,j)$ with a Gaussian Basis Kernel function followed by a MLP:

$$\Phi_{i,j}^{\text{energy}} = \text{MLP}(\phi_{\text{energy}}(\mathbf{F}(i,j))), \quad \Phi_{i,j}^{\text{dist}} = \text{MLP}(\phi_{\text{dist}}(\mathbf{D}(i,j))), \tag{3}$$

where $\phi_{\text{energy}}$ and $\phi_{\text{dist}}$ denote the learnable Gaussian Basis Kernel function that can map energy frustration score and distance score to a $d$-dimensional vector (See Appendix C for more details). MLP is further deployed to transform these vectors to the space of attention scores. We then add the resulting $\Phi_{(i,j)}^{\text{dist}}$ and $\Phi_{(i,j)}^{\text{energy}}$ as bias terms to the self-attention mechanism. Denote $\mathbf{A}_{i,j}^k$ as the $(i,j)$-element of the Query-Key product matrix in $k$-th attention layer, we have:

$$\begin{aligned} \mathbf{A}_{i,j}^k &= \frac{(\mathbf{h}_i^{k-1}\mathbf{W}_Q)(\mathbf{h}_j^{k-1}\mathbf{W}_K)^T}{d} + \Phi_{i,j}^{\text{energy}} + \Phi_{i,j}^{\text{dist}} \\ \mathbf{H}^k &= \text{softmax}(\mathbf{A}^k)\mathbf{H}^{k-1}\mathbf{W}_V, \end{aligned} \tag{4}$$

where $\mathbf{H}^k \in \mathbb{R}^{N \times d}$ denotes the updated representation matrix. And $\mathbf{W}_Q$, $\mathbf{W}_K$, and $\mathbf{W}_V$ are projection matrices for the query, key, and value transformations.

## 3.2 Enhancing Enzyme Representation Learning with Active Site Knowledge

To incorporate crucial active-site knowledge into enzyme representation learning, we use three strategies: (i) an auxiliary active-site prediction task to strengthen the enzyme encoder, (ii) large-scale pretraining for active-site prediction to capture general catalytic patterns, and (iii) an active site-aware pooling mechanism that emphasizes catalysis-related residues. Next, we give more details.

**Active Site Prediction.** Active sites play a key role in catalyzing the protein cleavage. Hence, active-site information can provide essential understandings of enzyme functions. As a result, we deploy the active site prediction as the auxiliary task to benefit the enzyme encoder training by:

$$\hat{a}_i = \text{sigmoid}(\mathbf{h}_i \cdot \mathbf{w}_a), \tag{5}$$

where $\hat{a}_i \in [0, 1]$ denotes the probability of the $i$-th residue being the active site, $\mathbf{h}_i \in \mathbb{R}^d$ is the representation of $i$-th residues in the enzyme, and $\mathbf{w}_a \in \mathbb{R}^d$ denotes the learnable parameters for active site prediction.

**Pretraining with the Supplemented Enzyme Set $\mathcal{D}_a$.** The number of enzymes annotated in cleavage site database is limited, which poses a significant challenge for the effective training of enzyme encoders [27]. Despite the limited enzyme coverage in cleavage site database, abundant enzymes are annotated with active sites. Therefore, we select enzymes highly homologous to the target proteolytic enzymes in biological function to expand the pretraining dataset. Formally, the objective function of pretraining the enzyme encoder on the supplemented enzyme set $\mathcal{D}_a$ can be written as:

$$\min_{\theta_e, \mathbf{w}_a} \mathcal{L}_a(\mathcal{D}_a) = \frac{1}{|\mathcal{D}_a|} \sum_{(\mathcal{P}^e, \mathbf{a}) \in \mathcal{D}_a} l_{\text{BCE}}(\hat{\mathbf{a}}, \mathbf{a}), \tag{6}$$

where $\theta_e$ denotes the parameters of enzyme encoder. $\hat{\mathbf{a}} = [\hat{a}_1, ... \hat{a}_N]$ denotes the probability vector of active site on the enzyme $\mathcal{P}^e$. $l_{\text{BCE}}$ denotes the element-wise binary cross entropy loss. By pretraining on a large corpus of enzyme sequences, we allow the model to capture broader structural and functional patterns common across enzymes.

**Active Site-Aware Pooling.** To obtain enzyme representation from a sequence of residue representations, a pooling operation such as mean pooling is required. However, residues that are active sites are more critical for enzymatic activity. Intuitively, these active sites should contribute more in the aggregated enzyme representation [28]. Therefore, we design an active site-aware pooling mechanism, whose pooling weights are based on the predicted active site probabilities. Let $\hat{a}_i \in \mathbb{R}^N$ be the predicted probability that $i$-th residue is an active site in an $N$-residue enzyme. The active site-aware pooling can be written as:

$$\mathbf{h}^e = \text{softmax}([w_1, \ldots, w_N])\mathbf{H}, \quad w_i = f(\hat{a}_i), \tag{7}$$

where $\mathbf{H} \in \mathbb{R}^{N \times d}$ is residue representation matrix from by the enzyme encoder. $f(\cdot)$ is a learnable function that will map each $\hat{a}_i$ to the pooling weight $w_i$, see in appendix C. With the active site-aware pooling, we would be able to encourages the model to focus on catalytically relevant segments of the enzyme.

## 3.3 Cleavage Site Prediction

**Substrate Protein Encoding.** As annotations tying substrate residues to intrinsic energetic states are unavailable, we omit the energetic frustration of the substrate protein. And we only input residue feature matrix $\mathbf{X}^s$ and distance matrix $\mathbf{D}^s$ of the substrate $\mathcal{P}^s$ are integrated in the transformer: $\mathbf{H}^s = \text{Transformer}(\mathbf{X}^s, \mathbf{D}^s)$, where $\mathbf{H}^s \in \mathbb{R}^{|\mathcal{P}^s| \times d}$ is the substrate representation. Further details are provided in Appendix C.

**Cleavage Site Prediction.** During protein hydrolysis, enzymes generally recognize local residue sequences about 15–30 residues in length. To reflect this biological behavior, we predict whether a subsequence of length $l$ in substrate $\mathcal{P}^e$ will be cleaved by enzyme $\mathcal{P}^e$. Formally, this process can be written by:

$$\hat{c}_t^{e,s} = \text{MLP}(\text{CONCAT}(\mathbf{H}_{t:t+l}^s, \mathbf{h}^e)) \tag{8}$$

where $\mathbf{H}_{t:t+l}^s$ is a contiguous slice taken directly from the substrate representation matrix $\mathbf{H}^s$ and $\mathbf{h}^e$ is the enzyme representation obtained by Eq.(7). The length of the subsequence is set as 31 (15

residues on each side.). The optimization function of cleavage site prediction can be written as:

$$\mathcal{L}_c(\mathcal{D}_c) = \frac{1}{|\mathcal{D}_c|} \sum_{(\mathcal{P}^e, \mathcal{P}^s, \mathbf{c}^{e,s}) \in \mathcal{D}_c} l_{\text{BCE}}(\mathbf{c}^{e,s}, \hat{\mathbf{c}}^{e,s}) \tag{9}$$

where $\hat{\mathbf{c}}^{e,s} = [\hat{c}_1^{e,s}, \ldots, \hat{c}_{|\mathcal{P}^s|}^{e,s}]$ denotes the probability vector of cleavage site within the substrate $\mathcal{P}^s$ given the enzyme $\mathcal{P}^e$. $l_{\text{BCE}}$ is the element-wise binary cross entropy loss.

### 3.4 Final Objective Function

For each enzyme $\mathcal{P}^e \in \mathcal{D}_c$ in the cleavage site database, their active sites are also included in the $\mathcal{D}_a$. Consequently, we combine the cleavage site prediction loss and the active site prediction to jointly train the whole framework by:

$$\min_{\theta} \mathcal{L}_c(\mathcal{D}_c) + \lambda \mathcal{L}_a(\mathcal{D}_a^c), \tag{10}$$

where $\theta$ denotes all parameters in UniZyme including the enzyme encoder, substrate encoder, active prediction module and cleavage site prediction module. $\mathcal{D}_a^c \subset \mathcal{D}_a$ provides the active-site annotations for the enzymes in $\mathcal{D}_c$.

## 4 Experiments

In this section, we conduct experiments to answer the following research questions:

- **RQ1**: How does the UniZyme perform in supervised cleavage site prediction?
- **RQ2**: How well does UniZyme generalize to cleavage site prediction for zero-shot enzymes?
- **RQ3**: How do the design of biochemically-informed enzyme encoder and utilization of active-site knowledge contribute to the performance of UniZyme?

### 4.1 Experimental Setup

**Dataset.** The cleavage site dataset $\mathcal{D}_c$ is sourced from the **MEROPS** database, which provides annotations for roughly 10k substrate proteins across 876 enzymes. With a standard dataset expansion procedure commonly used in cleavage site prediction[10, 18, 20, 21], which propagates substrate-site annotations across enzymes within the same family, we obtain 220k valid enzyme–substrate pairs. The supplemented dataset $\mathcal{D}_a$ is constructed by combining enzyme active-site annotations in **MEROPS** [22] and **UniProt** [23]. Specifically, MEROPS provides active sites for the enzymes already included in $\mathcal{D}_c$. UniProt provides hydrolase enzymes with the EC number of 3.4.*.* that are highly homologous to the proteolytic enzymes in $\mathcal{D}_c$, resulting to 11,530 enzymes with active sites.

**Evaluation.** To demonstrate the generalization ability of UniZyme, we evaluate its performance on protein cleavage site prediction for both seen enzymes (supervised) and unseen enzymes (zero-shot). The dataset split of the MEROPS for supervised and zero-shot setting are in Tab. 2. The dataset construction details can be found in Appendix A.

Table 2: Splits of MEROPS for evaluation.

|  | Training | Supervised Test | Zero-Shot Test |
|---|---|---|---|
| Enzyme Families | 677 | 69 | 23 |
| Substrate-Enzyme Pairs | 197,613 | 20,360 | 5,345 |

- **Supervised Setting**: In this scenario, target enzymes are paired with novel substrates that were not present in the training data. We restrict our evaluation to enzyme families with at least five unique substrates, yielding 69 enzyme families and a total of 21k enzyme–substrate pairs. For each family, we randomly split the data 70/10/20 into training, validation, and test sets.
- **Zero-shot Setting**: In the zero-shot setting, target enzymes are entirely held out during both training and pretraining. We consider only families with at least five distinct enzymes, reserving 20% of each family's enzymes as a test set. In total, this yields 23 enzyme families (5.3k enzyme–substrate pairs), with all test enzymes sharing under 60% sequence identity with any enzyme seen during training or pretraining.

**Baseline Methods.** To evaluate our model, we compare with the three categories of baselines. (i) *Specialized models*: **CAT3** [9] and **ScreenCap3** [18] are two models specifically designed for

C14.003 enzyme family. (ii) *Deep models for an individual enzyme*: **ProsperousPlus** [10], Deep-Cleave [20],and **DeepDigest** [29] are state-of-the-art deep learning methods focusing an individual enzyme system. To compare with UniZyme across multiple enzyme families, multiple predictors are trained for each baseline method, where each predictor is trained with data of an individual enzyme. Thus, these baselines are limited to supervised setting where an enzyme is provided with substantial experimental data. (iii) *Models revising UniZyme with baseline enzyme encoders*: Research on cleavage site prediction for zero-shot enzymes remains limited. The closest works are enzyme-substrate reaction predictors, namely **ClipZyme** [30] and **ReactZyme** [31], which encode both enzymes and substrates for reaction prediction. However, these models operate at the substrate level and cannot directly predict specific cleavage sites. Therefore, to demonstrate the superiority of the proposed enzyme encoder and enhancement strategies in Unizyme, we substitute the Unizyme's enzyme encoder with enzyme encoders from ClipZyme and ReactZyme. This yields two baselines for comparison in both supervised and zero-shot settings. For ClipZyme, we utilize their EGNN enzyme encoder pretrained for enzyme-substrate reaction prediction. See Appendix B for more details of baselines.

**Implementation Details.** We utilized the esm2-t12-35M-UR50D model to generate 480-dimensional residue features for both enzymes and substrates. The hyperparameter $\lambda$ is selected based on the validation set under supervised setting. Hyperparameter analysis are in Sec 4.5. Each experiment is conducted with 5 runs with different random seeds. To ensure a fair evaluation, hyperparameters of trainable baselines were selected by validation set. More details are in Appendix C.

## 4.2 Supervised Cleavage Site Prediction

To answer **RQ1**, we compare our UniZyme with various existing methods in supervised cleavage site prediction, ensuring that all models are trained and evaluated on the same data. As mentioned in Sec. 4.1, baselines designed for individual enzymes require training a separate predictor for each enzyme family. ClipZyme and ReactZyme are baselines revising our UniZyme with their enzyme encoders, avoiding repeated training. The overall comparisons on all 69 enzymes in supervised setting are given in Tab. 3 and Fig. 3. Results on 8 representa-

Table 3: Performance comparisons on overall 69 enzyme families under supervised setting.

| Model | Average PR-AUC (%) | Rate of Rank 1 (%) | Average Rank |
|---|---|---|---|
| UniZyme | **79.3** | **75.0** | **1.26** |
| ReactZyme | 70.0 | 7.5 | 2.51 |
| ClipZyme | 74.7 | 17.5 | 1.91 |
| ProsperousPlus | 7.2 | 0.0 | 4.65 |
| DeepCleave | 6.5 | 0.0 | 4.61 |
| DeepDigest | 4.2 | 0.0 | 5.52 |

tive enzymes are given in Tab. 4. Specifically, we can observe that: (**i**) Methods focusing on a single enzyme generally show poor performance; whereas those trained on multiple enzymes such as Our UniZyme and ReactZyme achieve significantly better results. This highlights the advantage of developing a unified cleavage site predictor across diverse proteolytic enzymes. (**ii**) Compared with ClipZyme which adopts an enzyme encoder pretrained with the enzyme-substrate reaction task, the proposed UniZyme achieves much better performance. It implies the effectiveness of active-site information in enhancing the enzyme encoder. (**iii**) Our UniZyme also consistently outperforms the ReactZyme by a large margin. This is because of the deployment of biochemically-informed enzyme encoder and the active-site knowledge.

Table 4: PR-AUC (%) on 8 out of 69 enzyme families under supervised setting. Note that ScreenCap3 and CAT3 are specialized models for the C14.003 enzyme family.

| Model | C01.034 | C14.003 | C14.005 | M13.001 | M24.026 | S01.001 | S01.224 | S08.070 |
|---|---|---|---|---|---|---|---|---|
| UniZyme | **78.1±1.3** | **45.9±1.2** | **52.2±0.9** | **60.1±1.1** | **87.2±0.8** | **82.1±1.2** | **62.4±1.1** | **85.0±1.3** |
| ReactZyme | 64.4±0.7 | 43.8±0.8 | 47.6±0.9 | 48.0±0.7 | 70.3±1.4 | 69.6±1.2 | 27.0±0.9 | 66.0±0.8 |
| ClipZyme | 71.1±0.7 | 35.3±0.9 | 43.2±1.0 | 37.1±1.3 | 73.1±0.8 | 73.8±0.9 | 41.7±0.6 | 73.8±0.9 |
| DeepDigest | 3.2±0.5 | 0.5±1.1 | 2.4±1.1 | 16.7±1.3 | 1.4±0.4 | 21.1±0.9 | 1.2±1.0 | 2.2±1.2 |
| DeepCleave | 4.8±1.1 | 1.0±1.1 | 4.2±0.9 | 19.0±1.0 | 7.1±1.2 | 26.9±1.4 | 5.0±0.5 | 18.3±0.6 |
| ProsperousPlus | 4.6±1.3 | 26.6±2.1 | 15.9±0.5 | 21.1±0.8 | 2.4±0.8 | 16.3±1.0 | 4.3±0.9 | 43.6±1.0 |
| CAT3 | – | 18.5±6.2 | – | – | – | – | – | – |
| ScreenCap3 | – | 29.2±16.0 | – | – | – | – | – | – |

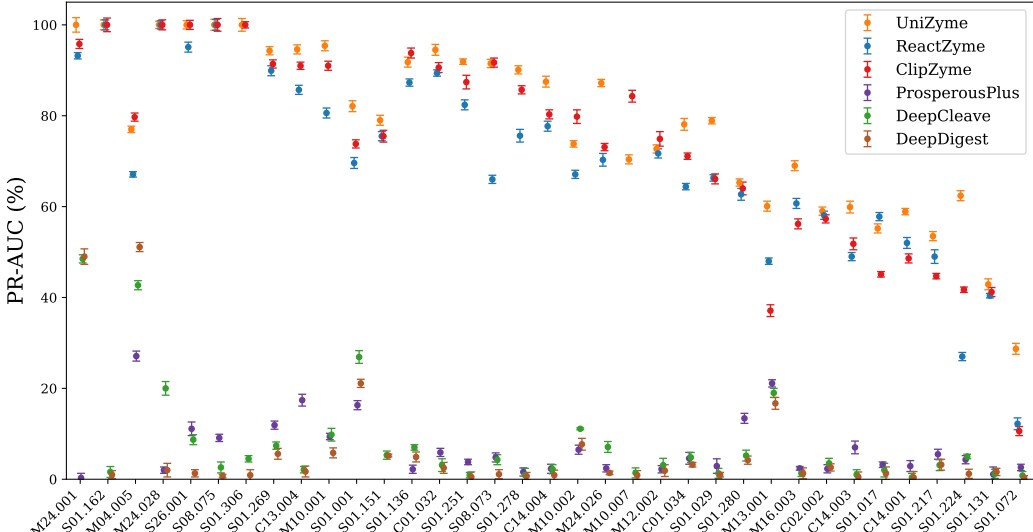

Figure 3: Per-family PR-AUC (%) across 69 supervised enzymes (Part 1: Enzymes 1–35). Results for the remaining 34 enzymes are provided in Appendix M (Fig. 9).

### 4.3 Cleavage Site Prediction for Zero-Shot Enzymes

To answer **RQ2**, we evaluate the performance of UniZyme on the zero-shot benchmarks, where enzymes are unseen during the training/pretraining phase. We adopt Needleman–Wunsch algorithm to ensure all zero-shot enzymes have under 60% sequence similarity with any enzyme used for training and pretraining. Since enzyme-specific models cannot handle novel enzymes, we only compare UniZyme to ReactZyme and ClipZyme, both of which are modified to predict cleavage sites for novel enzymes. The summarized results are given in Tab. 5, 6 and Fig. 5. From the results, we can observe that our UniZyme consistently outperform the baseline methods. In particular, UniZyme exceeds ReactZyme and ClipZyme by more than 7% in PR-AUC across most enzyme families. This improvement stems from the utilization of active-site knowledge in the enzyme modeling and the biochemical-informed encoder, promoting the generalization ability of UniZyme to unseen enzymes.

To further demonstrate the generalization ability of our model on unseen enzymes, we applied it to identify potential HIV-1 enzyme substrates and predict their cleavage sites as shown in Fig. 4. We present the model's prediction on an unseen HIV-1 enzyme acting on a 147-residue substrate (P62157) with four experimentally validated HIV-1 cleavage sites. As shown in Fig. 4, UniZyme can accurately predict the four annotated cleavage sites, achieving 100% accuracy. Additionally, UniZyme successfully predicts the cleavage sites of a test case (Uniprot: P00698). This demonstrates the ability to analyze cleavage sites for any potential protein of HIV-1 enzymes. This result provides valuable insights for therapeutic intervention and the development of inhibitors targeting HIV-1 enzymes.

Table 5: PR-AUC (%) on 8 out of 23 zero-shot enzyme families.

| Model | A01.009 | C01.060 | C02.002 | M10.001 | M10.004 | M12.217 | S01.010 | S01.217 |
|---|---|---|---|---|---|---|---|---|
| UniZyme | **37.5±0.6** | **84.3±1.3** | **66.4±0.6** | **81.1±1.0** | **82.8±1.8** | **93.2±1.3** | **61.0±0.8** | **65.0±1.2** |
| ReactZyme | 18.0±0.3 | 62.7±0.8 | 49.7±1.2 | 76.0±1.2 | 71.0±2.8 | 75.0±1.1 | 23.0±1.1 | 42.9±0.8 |
| ClipZyme | 25.2±0.6 | 59.2±0.9 | 48.6±1.2 | 76.8±0.6 | 56.5±3.5 | 84.4±0.9 | 18.9±0.9 | 41.3±0.6 |

Table 6: Performance comparisons on overall 23 enzyme families under zero-shot setting.

| Model | Average PR-AUC (%) | Rate of Rank 1 (%) | Average Rank |
|---|---|---|---|
| UniZyme | **71.1** | **78.3** | **1.30** |
| ReactZyme | 64.7 | 21.7 | 2.04 |
| ClipZyme | 61.7 | 0.0 | 2.65 |

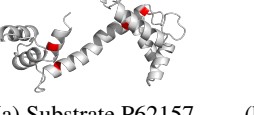

(a) Substrate P62157 (Accuracy = 1.0)

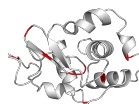

(b) Substrate P00698 (No Ground Truth)

Figure 4: Predicted HIV-1 enzyme cleavage sites highlighted in red (threshold = 0.5).

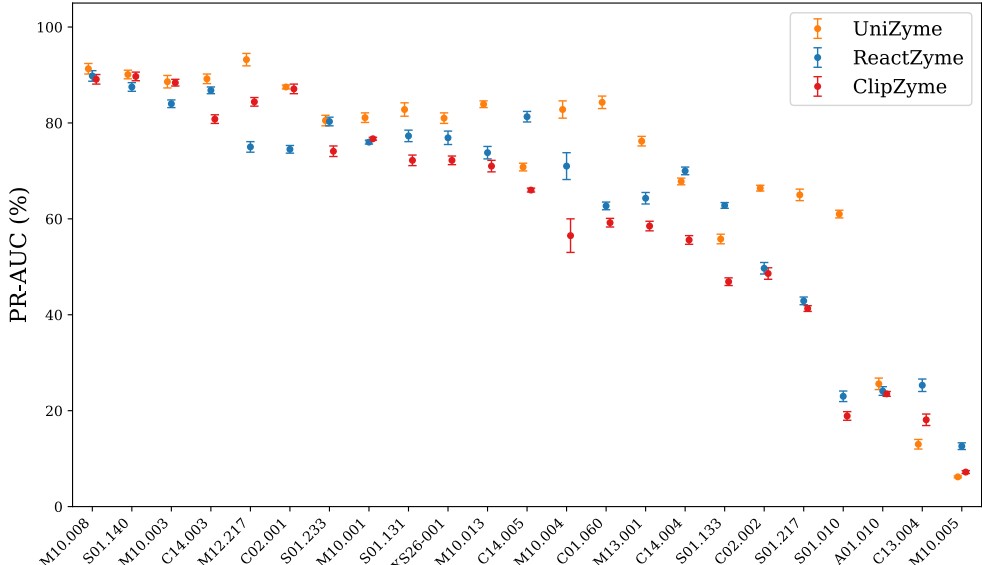

Figure 5: PR-AUC (%) on the zero-shot benchmark.

## 4.4 Ablation Studies

To answer **RQ3**, we conduct a series of ablation studies to understand the contributions of biochemically-informed enzyme encoder and the active-site knowledge. To demonstrate the effectiveness of the biochemically-informed enzyme encoder, we remove energy frustration and 3D structure, resulting in a variant named **UniZyme\SE**. To show the benefits brought by pretraining on general enzymes with active site prediction, we trained a variant, **UniZyme\P**, which excludes the enzyme pretraining phase. To further demonstrate the enhancement of active-site knowledge to the model, we remove the active site prediction in the pretraining/training phase. Additionally, the active site-aware pooling is replaced with average pooling, resulting in a variant named **UniZyme\A**. Fig. 6 shows the PR-AUC scores across different enzyme families in both zero-shot and supervised settings. More details can be found in the Tab. 17. From these results, we observe:

- UniZyme consistently achieves better results than UniZyme\SE. This indicates that the incorporation of structural-energy features in the biochemically-informed enzyme encoder can enable stronger generalization and performance in cleavage site prediction.
- UniZyme\P significantly performs worse than UniZyme on both supervised and zero-shot setting. This verifies that pretraining on a supplemented enzyme set with active site prediction can produce a more transferable enzyme encoder for cleavage site prediction.
- UniZyme outperforms UniZyme\A by a large margin. This demonstrates that the active-site knowledge can enhance the enzyme-catalyzed cleavage site prediction.

## 4.5 Hyperparameter Analysis

In this subsection, we investigate how the hyperparameter $\lambda$ affects the UniZyme. $\lambda$ controls to contribution of active site prediction loss to the training of UniZyme. To explore the hyperparameter analysis, we vary $\lambda$ as {100, 10, 1, 0.1, 0.01} in the training phase of UniZyme. Due to the expensive computational cost in training on the full dataset $\mathcal{D}_c$, we conduct the hyperparameter analysis with 3% of training data in various enzyme families. Performance on these enzymes are given in Fig. 7. We can find that while $\lambda = 100$ produces competitive ROC-AUC results, it leads to suboptimal PR-AUC. Small values like 0.1 and 0.01 cause a noticeable drop in ROC AUC (e.g. C14.005). Among the tested values, $\lambda = 10$ demonstrates the most consistent performance, achieving strong PR-AUC (e.g., M10.004) while maintaining competitive ROC AUC across datasets such as C14.003 and A01.009. Thus, we selected $\lambda = 10$ as the optimal choice for final training.

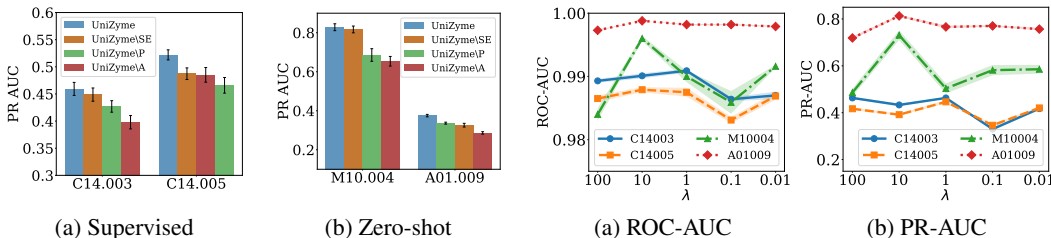

| (a) Supervised | (b) Zero-shot | (a) ROC-AUC | (b) PR-AUC |

Figure 6: Ablation studies of UniZyme.     Figure 7: Hyperparameter sensitivity analysis.

# 5 Related Works

**Protein Representation Learning**. Protein representation learning aims to effectively capture and represent the structural and functional features of proteins for downstream tasks. Inspired by large language models, recent years have seen the emergence of sequence-based pre-trained models such as ESM [32] and ProtTrans [33]. In terms of methods that utilize structural information, geometric graph neural networks [34–38] and transformers with structural constraints [26, 39, 40] have become widely used architectures. These approaches show that structural pretraining can significantly benefit downstream performance.

Beyond individual model designs, systematic benchmarks have begun to evaluate how different pretraining paradigms transfer to real-world applications. In particular, **Protap** [41] establishes a comprehensive benchmark that jointly compares backbone architectures, pretraining objectives, and domain-specific models across both general and specialized protein tasks. It covers five representative applications—including enzyme-catalyzed cleavage site prediction and PROTAC-mediated degradation—and provides unified data splits, evaluation protocols, and analysis of how structural inductive biases interact with pretraining scale. Within this framework, UniZyme is integrated as the representative enzyme–substrate model, demonstrating how incorporating biochemical priors such as energy frustration and active-site cues can complement large-scale foundation encoders.

**Cleavage Site Prediction.** Early prediction of enzyme-catalyzed cleavage sites relied on substrate sequence motifs, such as CAT3 [9] and Screen-Cap3 [18]. Subsequent works including Procleave [21] and ProsperousPlus [10] began to integrate substrate structural features to better capture enzymatic preferences. Deep learning approaches such as DeepCleave [20], DeepDigest [29], and DeepNeuropePred [17] further advanced the field by leveraging CNNs and protein language models. However, these are enzyme-specific models, applicable only to individual targets and often ignoring active-site information. Building on these developments and contextualized by the Protap benchmark, we propose a unified cleavage-site predictor enhanced with explicit active-site knowledge.

# 6 Conclusion and Future work

In this paper, we study a novel problem of developing a unified protein cleavage site predictor for diverse proteolytic enzymes. Specifically, we design a biochemically-informed enzyme encoder and incorporate redundant enzyme active-site information. Our experimental results demonstrate that UniZyme outperforms baselines by a large margin across various enzyme-substrate families, particularly excelling in zero-shot scenarios. Ablation studies further demonstrate the effectiveness of each proposed module in UniZyme. There are two directions that need further investigation. First, while this study focuses on proteolytic enzymes, we will extend to other categories of enzymes and substrates, and investigate whether enzyme-catalyzed reactions follow scaling law. Second, if more hydrolysis process data becomes available, incorporating dynamic structural information may improve prediction accuracy.

# 7 Acknowledgment

This material is based upon work supported by, or in part by, the National Natural Science Foundation of China (NSFC) under grant number 62506316. The findings in this paper do not necessarily reflect the view of the funding agencies.

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

**Algorithm 1** Training algorithm of UniZyme

---

**Require:** Supplemented enzyme set $\mathcal{D}_a$, Cleavage site prediction dataset $\mathcal{D}_c$, hyperparameters $\lambda$
**Ensure:** A unified cleavage site predictor $f_\theta$
 1: Initialize features of enzyme and substrate protein by ESM-2
 2: Pretrain enzyme encoder on $\mathcal{D}_a$ with $\mathcal{L}_a$ by Eq.(6)
 3: **for** epoch = 1 to $N$ **do**
 4:     **for** each batch $(\mathcal{P}^e, \mathcal{P}^s)$ in $\mathcal{D}_c$ **do**
 5:         Compute the distance matrix $\mathbf{D}^e$, and energetic frustration matrix $\mathbf{F}^e$ from enzyme structure
           by Eq.(2)
 6:         Encode the enzyme and substrate protein by Eq.(4)
 7:         Obtain the enzyme representation with active site-aware pooling by Eq.(7)
 8:         Predict active sites of enzymes by Eq.(5)
 9:         Predict cleavage sites of substrate proteins by Eq.(8)
10:         Update $\theta$ via $\nabla(\mathcal{L}_c + \lambda\mathcal{L}_a)$
11:     **end for**
12:     **if** validation loss increases for 3 epochs **then**
13:         **break**
14:     **end if**
15: **end for**

---

# A  Details of Data Curation and Benchmark Construction

## A.1  Data Curation and Preprocessing

For the cleavage dataset, we downloaded enzyme-substrate pairs from the MEROPS [22] database, collected substrate sequences from the UniProt database, and retrieved enzyme sequences recorded in MEROPS. Additionally, we compared the enzyme sequences between MEROPS and UniProt, excluding those with discrepancies, as such inconsistencies often result from asynchronous updates. To maintain controllable sequence lengths, we filtered out all enzyme and substrate sequences exceeding 1,500 residues.

Regarding the supplemented enzyme set with active sets, we first searched in the UniProt [23] database for enzymes with EC numbers starting with 3.4.*.* and filtered for reviewed data. Then, we selected entries with annotated active sites as our pretraining dataset. In addition, proteolytic enzymes in MEROPS are all annotated with active sites, and are combined as the supplemented enzyme set.

Protein structures were collected from the PDB [42] and AlphaFoldDB [43]. For proteins without available structures in these databases, we generated their structures using OmegaFold [44].

Additionally, all datasets are used in accordance with their respective licenses: MEROPS is distributed under the GNU Lesser General Public License (LGPL), UniProt under Creative Commons Attribution 4.0 (CC BY 4.0), the Protein Data Bank under CC0, AlphaFold DB under CC BY 4.0, and OmegaFold under the MIT License.

## A.2  Data Expansion

The MEROPS database classifies enzymes into categories based on their substrate cleavage sites. Enzymes belonging to the same MEROPS category typically share highly similar cleavage-site characteristics[22]. Drawing on previous work, we assume that minor sequence differences among enzymes of the same category can be disregarded. Consequently, the hydrolysis information from a substrate–enzyme pair is extended to all enzymes in that category.

Therefore, we expanded our dataset by matching each substrate not only with the originally mapped enzyme but also with other enzymes in the same MEROPS category. Through this procedure, we obtained approximately 220K valid enzyme–substrate pairings, involving 677 unique enzymes. Detailed distributions of enzyme and substrate pairs are provided in Tab. 7.

Table 7: Dataset statistics of training datasets.

| Utilization | Datasets | # Substrate-Enzyme Pairs | Enzymes | Substrates |
|---|---|---|---|---|
| Active site dataset | UniProt | NA | 11,530 | NA |
| Cleavage site dataset | MEROPS | 197,613 | 677 | 7,475 |

### A.3 Construction of Supervised and Zero-shot Benchmarks

**Supervised Setting.** We selected MEROPS enzyme families containing at least five distinct substrates, yielding 69 families and approximately 21K enzyme–substrate pairs. All pairs in each family were randomly split 70/10/20 into training, validation and test sets. To ensure the test substrates were sufficiently distinct from those in training, we collected all substrates per family and computed pairwise sequence similarity using the Needleman–Wunsch algorithm (BLOSUM62, gap opening penalty=10, gap extension penalty=0.5). Substrates exhibiting less than 50% similarity to any other were deemed independent, and 20% of independent substrates were sampled to form the final test set. This procedure evaluates model generalization on more divergent substrates within each MEROPS family.

**Zero-shot Setting.** Following the same selection criteria, we identified MEROPS families with at least five enzymes, yielding 23 families and roughly 5.3K enzyme–substrate pairs. In each family, 20% of the enzymes were randomly set aside as a zero-shot test set. To guarantee that these test enzymes were truly unseen, we computed pairwise sequence identities against all training and pretraining enzymes using the Needleman–Wunsch algorithm (BLOSUM62, gap opening penalty = 10, gap extension penalty = 0.5) and retained only those enzymes whose identity fell below a 60% threshold. This procedure prevents any overlap between zero-shot test enzymes and the training/pretraining pool, rigorously evaluating the model's ability to generalize to novel enzymes.

## B Details of Baselines

Below, we provide additional details on how we adapt, retrain, or utilize each baseline for comparison. Unless otherwise specified, all default hyperparameters are used as in the original implementations of these methods. For any required data, we convert our data format accordingly.

**ProsperousPlus** [10] and **DeepDigest** [29] all provide publicly available code, enabling us to retrain their models within our supervised setting. We use the same training and test sets as those used for our method, specifically for the supervised benchmark. We adopt the default training code from each repository while ensuring that all other settings remain consistent.

**ScreenCap3** [18] and **CAT3** [9], specialized for the C14.003 enzyme, do not provide publicly available datasets or source code for retraining. Instead, they each offer a prediction platform: a web server for ScreenCap3 and standalone software for CAT3. We use these platforms to generate predictions on our test set. Since their training data are not publicly accessible, we can only report their performance as is, with the caveat that neither model can be applied to other enzymes.

We also compare with two recent enzyme–substrate interaction models, **ClipZyme** [30] and **ReactZyme** [31], which were originally proposed for reaction rather than cleavage prediction. **ReactZyme** encodes enzymes with an ESM-2 plus MLP pipeline, but since its trained weights are unavailable, we retrain it from scratch on our dataset. **ClipZyme** employs an E(n) Equivariant Graph Neural Network (EGNN) to incorporate structural information into its enzyme encoder. Both models use average-pooling to aggregate the extracted enzyme features and are trained without activation-site loss. To highlight the effect of leveraging active-site knowledge, we keep the original pretrained EGNN for ClipZyme as is and integrate it into our cleavage-site prediction framework, adding only a linear projection layer to interface with the cleavage-site prediction module.

## C Implementation Details

**Framework and Hardware.** We implemented our models in PyTorch and trained using the Adam optimizer with a learning rate of $1 \times 10^{-4}$ and a batch size of 48. All experiments were conducted

on eight NVIDIA A6000 GPUs(48G). We adopted an early stopping strategy with a patience of 3 epochs, monitoring the validation loss to prevent overfitting.

**Substrate Representations.** Similar to the enzyme pipeline, but without energetic frustration, each residue is embedded by ESM-2 padded to 1500 length. We compute pairwise $C\alpha$-distances $\mathbf{D}^s(i,j) = \|\mathbf{r}_i - \mathbf{r}_j\|_2$, then applying a reciprocal transform. Each distance entry is processed by a Gaussian basis kernel and MLP, yielding a bias term $\Phi_{i,j}^{\text{dist}}$ added to the attention score:

$$\mathbf{A}_{i,j}^k = \frac{(\mathbf{h}_i^{k-1}\mathbf{W}_Q)(\mathbf{h}_j^{k-1}\mathbf{W}_K)^T}{d} + \Phi_{i,j}^{\text{dist}}, \tag{11}$$

thus incorporating structural information. The substrate representation $\mathbf{H}^s \in \mathbb{R}^{|\mathcal{P}^s| \times d}$ is obtained via

$$\mathbf{H}^s = \text{Transformer}(\mathbf{X}^s, \mathbf{D}^s), \tag{12}$$

with the same architecture as the enzyme encoder but omitting energy-related parameters.

**Pooling Weight Function.** In Active Site-Aware Pooling module, $f(\cdot)$ is a learnable mapping that transforms each predicted active-site probability into its final pooling weight, in direct analogy to how we use Gaussian kernels to map energy and distance into attention biases.

Concretely, we first pass the scalar probability $\hat{a}_i$ through a Gaussian basis expansion:

$$\boldsymbol{\phi}_{\text{act}}(\hat{a}_i) = \left[ \phi_{\text{act},1}(\hat{a}_i), \ldots, \phi_{\text{act},K}(\hat{a}_i) \right] \in \mathbb{R}^K, \tag{13}$$

which produces a richer, multi-dimensional embedding. We then apply an MLP to collapse this embedding to a single weight:

$$w_i = f(\hat{a}_i) = \text{MLP}(\boldsymbol{\phi}_{\text{act}}(\hat{a}_i)). \tag{14}$$

**Energy Frustration Calculation.** We computed residue-pair frustration using the **Frustratometer** tool [45] with AWSEM (Associative Water-mediated Structure and Energy Model) potentials [46], disabling electrostatic interactions ($k_{\text{electrostatics}} = 0$) and enforcing a minimum sequence separation of 12 residues between residue pairs. Specifically, for each pair of residues $(i,j)$ in enzyme $\mathcal{P}^e$, the actual interaction energy $\mathbf{E}(i,j)$ was extracted from the AWSEM potential. To capture local energetic fluctuations, we generated an ensemble of randomized configurations (where the sequence or side-chain identities are shuffled while preserving the protein backbone), thereby obtaining a distribution of interaction energies for each pair.

Let $\mu_{\text{rand}}(i,j)$ and $\sigma_{\text{rand}}(i,j)$ be the mean and standard deviation of these interaction energies over the randomized ensemble. The frustration score $\mathbf{F}(i,j)$ is then computed as:

$$\mathbf{F}(i,j) = \frac{\mathbf{E}(i,j) - \mu_{\text{rand}}(i,j)}{\sigma_{\text{rand}}(i,j)}. \tag{15}$$

A higher $\mathbf{F}(i,j)$ indicates that the local region around residues $(i,j)$ is more frustrated (i.e., further from minimal AWSEM-derived energy). Such regions often correspond to sites of functional importance in enzymes.

To estimate how $\mathbf{E}(i,j)$ deviates from an energetically minimal arrangement, we generated an ensemble of randomized "decoy" configurations for the same residue pair. These decoys preserve global geometry (e.g. backbone coordinates) but shuffle aspects such as side-chain packing or local environment, depending on the chosen protocol within the **Frustratometer**. Each decoy thus provides a distinct pairwise interaction energy. By sampling multiple decoys, we obtain an approximate distribution of energies $\tilde{E}_k(i,j)$, from which we compute:

$$\mu_{\text{rand}}(i,j) = \frac{1}{K}\sum_{k=1}^{K}\tilde{E}_k(i,j), \tag{16}$$

$$\sigma_{\text{rand}}(i,j) = \sqrt{\frac{1}{K-1}\sum_{k=1}^{K}\left(\tilde{E}_k(i,j) - \mu_{\text{rand}}(i,j)\right)^2}, \tag{17}$$

where $K$ is the number of randomized decoys (typically on the order of a few hundred in the **Frustratometer**).

**Gaussian Basis Kernel Function.** Following Transformer-M [26], we employ a set of learnable Gaussian basis kernels to transform a scalar input (e.g., the distance $\mathbf{D}(i,j)$ or the frustration score $\mathbf{F}(i,j)$) into a fixed-dimensional embedding. Concretely, suppose we have $K$ Gaussian kernels parameterized by $\{\mu^k, \sigma^k\}_{k=1}^K$. For an input scalar $x$, the Gaussian basis kernel function $\phi(x)$ is defined as:

$$\phi(x) = \left[\exp\left(-\tfrac{1}{2}\left(\tfrac{x-\mu^1}{\sigma^1}\right)^2\right),\ \exp\left(-\tfrac{1}{2}\left(\tfrac{x-\mu^2}{\sigma^2}\right)^2\right),\ \ldots,\ \exp\left(-\tfrac{1}{2}\left(\tfrac{x-\mu^K}{\sigma^K}\right)^2\right)\right]^\top. \quad (18)$$

Each kernel center $\mu^k$ and width $\sigma^k$ is learnable, allowing the model to adaptively capture different regions of the input space. We apply this basis expansion to both $\mathbf{D}(i,j)$ and $\mathbf{F}(i,j)$, producing a $K$-dimensional vector for each pair $(i,j)$. An MLP then projects this kernel output into the space of attention biases. We set the number of Gaussian basis functions to $K = 10$, each parameterized by learnable centers $\mu^k$ and widths $\sigma^k$. Notably, we maintain *separate* sets of Gaussian parameters for the energy and structure channels, ensuring that the model can adaptively learn distinct representations for each.

**Training Algorithm.** Each sample's ESM-2 embeddings (padded to length 1500), along with distance and energy frustration matrices, are fed into our model to predict both active-site and cleavage-site residues. We use a weighted binary cross-entropy loss and optimize with Adam for up to 15 epochs, applying early stopping (patience = 3) based on validation loss.

# D    Comparison of Active-Site Prediction Module

To comprehensively evaluate the active-site prediction module of **UniZyme**, we compared it with two representative structure-based models that utilize enzyme structural and sequence information: **GraphEC** [47] and **NodeCoder** [48]. Both models were reimplemented using their official repositories and retrained on the same large-scale active-site prediction dataset employed by UniZyme. The dataset statistics are summarized in Table 8. The performance of different models is reported in Table 9. **UniZyme** achieves consistently superior results compared to the baselines.

Table 8: Data statistics for the active-site prediction task.

| Dataset | Number of Enzymes | Number of Active Sites |
|---|---|---|
| Training | 9220 | 24891 |
| Test | 2349 | 6459 |

Table 9: Comparison of active-site prediction performance.

| Model | AUROC (%) | AUPR (%) | Precision (%) | Recall (%) | F1 (%) |
|---|---|---|---|---|---|
| UniZyme | 89.5 | 35.1 | 65.3 | 45.6 | 53.7 |
| GraphEC | 80.3 | 28.0 | 52.1 | 38.7 | 44.4 |
| NodeCoder | 67.4 | 17.8 | 32.6 | 22.3 | 26.5 |
| Random Guess | 50.0 | 3.2 | 3.2 | 50.0 | 6.0 |

# E    Structural Source Sensitivity Analysis

To evaluate the robustness of **UniZyme** with respect to the source of structural data, we partitioned the test set into four quadrants based on whether the enzyme and substrate structures were obtained from experimental (natural) or predicted sources. Table 10 reports the PR-AUC results under both supervised and zero-shot settings.

Table 10: PR-AUC (%) of UniZyme under different structural-source combinations.

| Structure Source | Zero-shot PR-AUC (%) | Supervised PR-AUC (%) |
|---|---|---|
| Both Natural Structures (3%) | 72.2 | 80.3 |
| Natural Enzyme + Generated Substrate (7%) | 71.9 | 77.9 |
| Natural Substrate + Generated Enzyme (8%) | 70.5 | 81.9 |
| Both Generated Structures (82%) | 69.4 | 78.3 |

## F   Statistical Significance Testing

To confirm that the performance improvements of **UniZyme** over baseline models are statistically significant, we conducted two-sample t-tests on PR-AUC scores across enzyme families under both supervised and zero-shot settings. Table 11 summarizes the results.

Table 11: Two-sample t-test results comparing UniZyme with baseline models.

| Setting | Comparison | t-value | p-value |
|---|---|---|---|
| Supervised | UniZyme vs ReactZyme | 7.18 | 7.8e-10 |
| Supervised | UniZyme vs ClipZyme | 5.09 | 3.0e-06 |
| Zero-shot | UniZyme vs ReactZyme | 2.61 | 1.6e-02 |
| Zero-shot | UniZyme vs ClipZyme | 4.27 | 3.1e-04 |

## G   Cross-Task Transferability: EC Number Classification

To assess whether the enzyme encoder of **UniZyme** captures generalizable biochemical signals, we evaluated it on the EC number classification task for proteases (EC 3.4.*.*). The same dataset split used in the cleavage-site prediction task was reused here. Table 12 presents the results in terms of AUROC.

Table 12: Performance of enzyme encoders on EC number classification.

| Model | AUROC (%) |
|---|---|
| UniZyme | 94.1 |
| ClipZyme | 90.2 |
| ReactZyme | 82.3 |

## H   Interpretability and Mechanistic Consistency

We conducted interpretability analyses to understand how **UniZyme** utilizes active-site information for cleavage-site prediction. Higher predicted active-site confidence consistently corresponds to better PR-AUC, indicating that UniZyme effectively leverages active-site cues (Table 13). Gradient-based attribution analysis further shows that active-site residues contribute more strongly to cleavage prediction than background residues (Table 14), confirming that the model's attention is aligned with catalytic regions. Moreover, perturbation experiments demonstrate that masking top predicted active-site residues causes a substantial PR-AUC drop, whereas perturbing random residues has minimal effect (Table 15), verifying that UniZyme's predictions are causally linked to the identified catalytic sites.

**Mechanistic Basis of Enzyme Cleavage.**   During protein hydrolysis, enzyme active sites provide a specific geometric and electrochemical environment that enables cleavage only at substrate residues exhibiting optimal complementarity. Thus, active-site geometry and chemistry directly influence cleavage-site specificity. Structural biology and mutational evidence strongly support this relationship: a single residue change in the S1 pocket of canonical serine proteases (e.g., trypsin vs. chymotrypsin)

can dramatically alter specificity by reshaping charge preference and side-chain accommodation [49, 50]. Loop variations near the active-site pocket can also reshape neighboring subsites and modulate substrate scope [51]. Furthermore, structural studies of Alzheimer's $\gamma$-secretase show that the architecture of the binding cleft and distal exosites critically determine substrate recognition, where mutations at the interface shift cleavage patterns [52]. Energetic and mutational scanning analyses of viral proteases (e.g., HCV NS3/4A) also reveal that substrates optimally filling the active-site groove undergo efficient catalysis, while suboptimal packing results in weak or absent cleavage [53].

These mechanistic observations provide biological grounding for UniZyme's interpretability analyses: the model's high sensitivity to predicted active-site residues mirrors the physicochemical principles underlying real enzymatic catalysis, reinforcing that UniZyme captures not only statistical correlations but also mechanistic causality.

Table 13: PR-AUC (%) across bins of predicted active-site confidence.

| Active-Site Probability Range | Zero-shot PR-AUC (%) | Supervised PR-AUC (%) |
|---|---|---|
| [0.8, 1.0] | 78.6 | 85.4 |
| [0.6, 0.8) | 73.8 | 80.3 |
| [0.4, 0.6) | 63.1 | 73.9 |
| [0.2, 0.4) | 57.7 | 61.6 |
| [0, 0.2) | 52.1 | 56.2 |

Table 14: Average attribution magnitude for active-site and background residues.

| Residue Type | Embed. Sens. (Sup.) | Upstream Attr. (Sup.) | Embed. Sens. (Zero-shot) | Upstream Attr. (Zero-shot) |
|---|---|---|---|---|
| Active-site residues | 0.68 | 0.74 | 0.55 | 0.60 |
| Background residues | 0.23 | 0.10 | 0.19 | 0.08 |

Table 15: Effect of perturbing pooling weights on PR-AUC (%).

| Perturbation Target | Sup. Orig. | Sup. Post | $\Delta$ (Sup.) | Zero Orig. | Zero Post | $\Delta$ (Zero) |
|---|---|---|---|---|---|---|
| Predicted active-site residues (Top-5) | 79.3 | 66.2 | 13.1 | 71.1 | 57.4 | 13.7 |
| Random non-active-site residues | 79.3 | 78.8 | 1.5 | 71.1 | 69.4 | 1.7 |

## I Computational Effectiveness

Tab. 16 reports wall-clock times measured on NVIDIA A6000 GPUs. Training was conducted on 8xA6000 GPUs; inference was profiled on a single A6000 GPU. As shown, UniZyme's end-to-end training cost is comparable to baselines, and its average per-pair inference latency remains within practical bounds (around 1s overhead).

Table 16: Training and inference times.

| Model | Total Training Time (h) | Inference per Pair (s) |
|---|---|---|
| Pretraining of UniZyme | 12.5 | — |
| UniZyme | 30.9 | 5.2 |
| ClipZyme | 31.4 | 3.5 |
| ReactZyme | 30.3 | 4.4 |

## J Ablation Studies

The Tab. 17 reports the average PR-AUC across 69 supervised families and 23 zero-shot families. Full UniZyme achieves the best results in both settings, and performance progressively declines when the SE, A, or P modules are ablated.

Table 17: Ablation Studies on 69 supervised and 23 zero-shot enzyme families.

| Model | Supervised | Zero-shot |
|---|---|---|
| **UniZyme** | 79.3±1.2 | 71.1±2.3 |
| **UniZyme\SE** | 75.3±1.9 | 65.3±1.4 |
| **UniZyme\A** | 72.5±1.5 | 60.6±1.8 |
| **UniZyme\P** | 73.0±2.2 | 62.1±2.7 |

## K Failure Case Analysis (Limitation)

In our supervised experiments, the M10.003 family consistently lagged behind C14.003 and C14.005, despite having comparable dataset sizes. To understand this, we computed the Shannon entropy of the amino acid distribution at positions P1–P6 around the cleavage site (P3–P4) in Tab. 18. Lower entropy indicates more conserved residues, which simplifies pattern recognition.

Table 18: Shannon entropy at positions around the cleavage site (P3–P4).

| Clan | P1 | P2 | P3 (cleavage site) | P4 (cleavage site) | P5 | P6 |
|---|---|---|---|---|---|---|
| C14.005 | 4.10 | 4.09 | **3.20** | 3.99 | 4.11 | 4.11 |
| C14.003 | 4.11 | 4.01 | **3.40** | 3.96 | 4.05 | 4.11 |
| M10.003 | 3.97 | 4.00 | 3.95 | 3.66 | 4.03 | 3.90 |

Both C14.003 and C14.005 exhibit markedly lower entropy at the cleavage-site residues (P3, P4), indicating conserved amino acids that aid substrate recognition. By contrast, M10.003 shows uniformly higher entropy, reflecting greater sequence diversity and fewer distinctive cleavage motifs. Moreover, as a metalloprotease, M10.003's activity depends on complex metal-ion coordination in its active-site, further complicating its cleavage specificity.

These observations suggest that M10.003's higher substrate diversity and more intricate catalytic mechanism underlie its lower prediction accuracy. In future work, we plan to integrate metal-ion binding data and more detailed active-site pocket features to better capture the unique determinants of metalloprotease specificity.

## L Broader Impacts

Accurately predicting protease-substrate cleavage sites across a wide enzymatic landscape is pivotal for therapeutic molecule design, industrial biocatalysis, and systematic studies of disease-associated proteolysis. By performing large-scale virtual mapping of these interactions, UniZyme enlarges the set of candidate targets that experimentalists can pursue, thereby accelerating lead prioritization in early-stage drug discovery, guiding the engineering of enzymes with enhanced specificity and stability, and enabling rapid evaluation of emerging viral proteases to bolster pandemic preparedness. Like any computational framework that substantially improves the efficiency of protease screening and functional prediction, UniZyme could be misapplied—for instance, to create proteases that undermine existing biologics or aid in the synthesis of harmful compounds.

## M Additional Experiments and Visualizations

As shown in Fig. 8, we conducted zero-shot testing on all enzymes not included in the training data to evaluate the model's capability in predicting enzyme active sites.

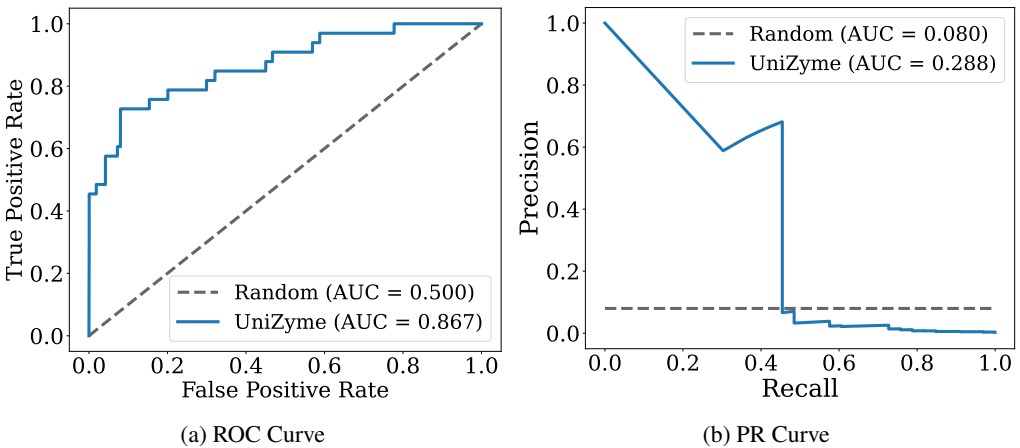

(a) ROC Curve

(b) PR Curve

Figure 8: Model performance of active-site prediction

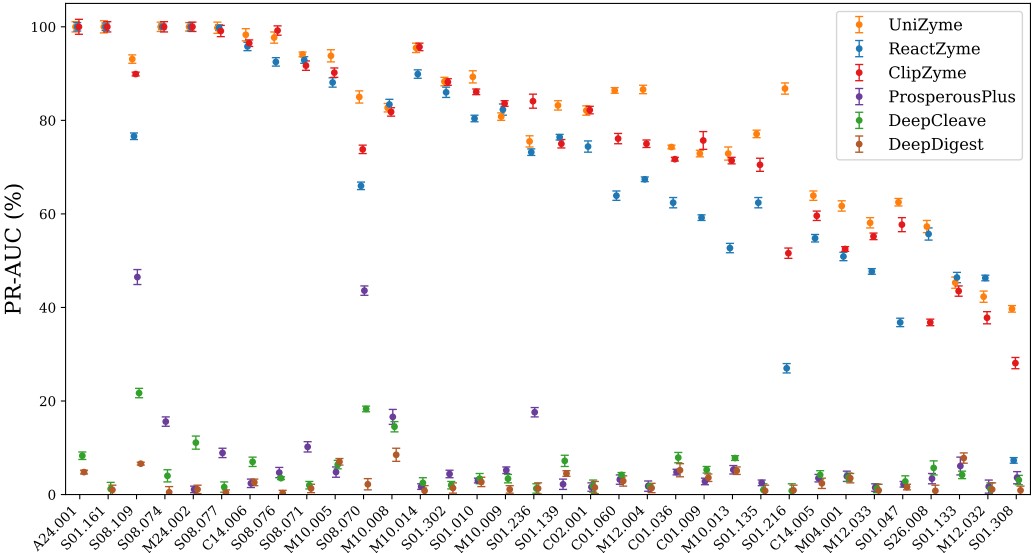

Figure 9: Per-family PR-AUC (%) across 69 supervised enzymes(Part 2: Enzymes 36–69) corresponding to Fig. 3.

