# OpenReview forum: "UniZyme: A Unified Protein Cleavage Site Predictor Enhanced with Enzyme Active-Site Knowledge"
_NeurIPS.cc/2025/Conference — NeurIPS 2025 poster_

### Official Review · Reviewer_8og7 · 2025-06-24

**Clarity:** 3
**Significance:** 3
**Originality:** 3
**Rating:** 4
**Confidence:** 4

**Summary:**

This paper proposes a unified protein cleavage site prediction model UniZyme, which aims to solve the problem that existing methods only target a single enzyme and lack generalization capabilities. The authors designed a framework that includes a biochemically aware enzyme encoder and an active site-aware pooling mechanism by integrating the active site knowledge and biochemical information of the enzyme, and enhanced the model performance through pre-training and joint training strategies. Experiments show that UniZyme significantly outperforms existing baselines in both supervised learning and zero-shot scenarios, especially in predicting the cleavage sites of unseen enzymes.

**Questions:**

1. Please provide some interpretable analysis of the relationship between the enzyme's active site and its cleavage site
2. Could you show whether length of enzymes has significant impact?

**Ethical Concerns:**

["NO or VERY MINOR ethics concerns only"]

**Final Justification:**

I would like to thank the authors for their detailed reply, addressing all my questions and providing extensive additional experiments to support the arguments. I would like to ask the authors to include all new experiments and explanations in the final version of their paper.

**Limitations:**

yes

**Quality:**

3

**Strengths And Weaknesses:**

Strengths:
1. Breaking the traditional single enzyme prediction model, a unified cross-enzyme prediction model based on active site knowledge is proposed for the first time, solving the pain point that existing methods cannot be generalized to new enzymes.
2. The key catalytic regions are captured by energy frustration score and active site-aware pooling, and the structural information is processed by Gaussian kernel function, which reflects the deep modeling of enzyme catalytic mechanism.
3. Comprehensive benchmarking is carried out to verify the performance in supervised (69 enzyme families) and zero-shot (23 enzyme families) scenarios. The PR-AUC indicators are significantly better than ClipZyme, ReactZyme and other baselines, especially in the zero-shot scenario, with an advantage of more than 7%.
4. Ablation experiments confirm the necessity of energy frustration (UniZyme\SE), pre-training (UniZyme\P) and active site pooling (UniZyme\A), and the mechanism analysis is rigorous.
5. Conduct failure case analysis, and analyze the limitations from the perspective of amino acid sequence entropy and catalytic mechanism for the poor prediction effect of the M10.003 family, reflecting the rigor of the research.
6. The paper has a clear logical structure, which progresses from the problem background, method design to experimental verification. The method section describes in detail the mathematical derivation and implementation details of modules such as enzyme encoders and active site pooling.

Weakness:
1. The framework diagram of the method is not clear enough, and the process is vague. It is recommended to put the active site prediction a between Enzyme Encoder and Active Site-Aware Pooling, which is more logical.
2. Limitations of enzyme encoder: The article mentions that "energetic frustration score calculation is based on AWSEM potential energy", but does not explain how to deal with the potential energy differences of different enzyme families (such as the energy characteristics of serine proteases and metalloprotease). It is recommended to supplement the adaptability analysis of specific enzyme families, or provide details of potential energy parameter settings in the appendix.
3. The results in the article are mainly presented in tabular form. It is recommended to display the distribution of the data for a more intuitive comparison.
4. The existing description does not explain the specific form of the pooling weight function f(.) (such as linear/nonlinear transformation). It is recommended to clarify whether the function is an activation function (such as ReLU) or a weighted mapping.
5. The article can provide the basis for the relationship between the active site and the cleavage site of the enzyme. It is recommended that the article can supplement the interpretable analysis of the model.

---

> ### Author Rebuttal · Authors · 2025-07-31
>
> Thanks for your valuable feedback. We address your questions in the following:
>
>
> ### **Response to clarity of framework diagram (Weakness 1)**
> Thank you very much for your valuable and practical suggestion. We will revise this diagram in the camera‑ready version.
>
>
> ### **Response to limitation of enzyme encoder (Weakness 2)**
> We use a single, consistent AWSEM parameter set for all enzyme families, as detailed in Appendix C and our GitHub repository. Specifically, we disable electrostatics (kelectrostatics = 0), enforce a 12‑residue sequence separation, generate 500 decoys by preserving backbone coordinates and randomizing side‑chains, and include the standard bond, angle, dihedral, contact, hydrogen‑bond, water‑mediated, and burial terms. We chose this uniform configuration because Freiberger et al. demonstrated that the same AWSEM coarse‑grained potential successfully identifies frustration patterns around active sites across diverse protease families [1].
>
>
>
> ### **Response to Data presentation (Weakness 3)**
>
> Thank you for the suggestion. In the camera‑ready version, we will update the figures and include more informative visualizations.
>
>
>
> ### **Response to pooling weight function f(.) (Weakness 4)**
> Here, f(.) is a learnable mapping that transforms each predicted active‑site probability into its final pooling weight, in direct analogy to how we use Gaussian kernels to map energy and distance into attention biases.
>
> Concretely, we first pass the scalar probability $\hat a_i$ through a Gaussian basis expansion:
>
> $$ \phi_{\mathrm{act}}(\hat a_i) = \bigl[\phi_{\mathrm{act},1}(\hat a_i), \dots, \phi_{\mathrm{act},K}(\hat a_i)\bigr] \in \mathbb{R}^K, $$
>
> which produces a richer, multi‑dimensional embedding. We then apply an MLP to collapse this embedding to a single weight:
>
> $$ w_i = f(\hat a_i) = \mathrm{MLP}\bigl(\phi_{\mathrm{act}}(\hat a_i)\bigr). $$
>
>
>
> ### **Response to interpretable analysis of the relationship between the enzyme's active site and its cleavage site (Weakness 5 and Question 1)**
>
> To address this, we provide an interpretability analysis that explores the relationship between predicted active-site probabilities and the quality of cleavage site prediction. Specifically, we aim to test whether higher-confidence active site signals are associated with more accurate cleavage predictions, thus demonstrating that the model indeed uses active-site information in a meaningful and interpretable way.
>
> Here we present the PR-AUC of cleavage site prediction across different ranges of predicted active site probabilities, aiming to illustrate the interpretability of how active site signals influence substrate cleavage site prediction. We categorize all test samples into five groups based on the average predicted active-site probability (of actual active sites) $\hat{a}_i$ of the enzyme. From the table below, we can find that cleavage prediction improves with higher predicted active site probabilities in both zero-shot and supervised settings, indicating that the model effectively leverages active site signals.
>
> | Avg. Active Site Prob. Range | Zero-shot Avg. PR-AUC (%) | Supervised Avg. PR-AUC (%) |
> |:----------------------------:|:----------------------:|:-----------------------:|
> | [0.8, 1.0] | 78.6 | 85.4 |
> | [0.6, 0.8) | 73.8 | 80.3 |
> | [0.4, 0.6) | 63.1 | 73.9 |
> | [0.2, 0.4) | 57.7 | 61.6 |
> | [0, 0.2) | 52.1 | 56.2 |
>
>
>
> ### **Response to impact of the length of enzymes (Question 2)**
>
> To assess whether enzyme length influences cleavage prediction performance, we grouped test enzymes by sequence length and reported the PR-AUC under both zero-shot and supervised settings.
>
>
> As shown in the table below, UniZyme maintains strong and stable performance across a wide range of lengths. While shorter enzymes ([0, 300)) achieve slightly higher accuracy, the model continues to perform robustly even for longer sequences, demonstrating its generalization ability across enzyme sizes.
>
>
> | **Enzyme Length Range (aa)** | **Zero-shot PR-AUC (%)** | | | **Supervised PR-AUC (%)** | | |
> |:---------------------------:|:--------------------------------------:|:------------------------------:|:------------------------------:|:-------------------------------------:|:------------------------------:|:------------------------------:|
> | | **UniZyme** | **ReactZyme** | **ClipZyme** | **UniZyme** | **ReactZyme** | **ClipZyme** |
> | [0, 300) | 73.0 | 65.4 | 63.5 | 82.9 | 74.6 | 77.2 |
> | [300, 600) | 68.1 | 60.2 | 62.5 | 75.7 | 68.3 | 70.4 |
> | [600, 900) | 70.6 | 61.7 | 58.2 | 82.5 | 72.1 | 75.6 |
> | [900, 1200) | 71.8 | 62.9 | 63.8 | 81.7 | 70.2 | 72.3 |
> | [1200, 1500] | 68.8 | 59.6 | 56.1 | 74.3 | 66.7 | 73.0 |
>
> ### **Reference**
> **[1] M.I. Freiberger, A.B. Guzovsky, P.G. Wolynes, R.G. Parra, & D.U. Ferreiro, Local frustration around enzyme active sites, Proc. Natl. Acad. Sci. U.S.A. 116 (10) 4037-4043**

---

> > ### Comment · Reviewer_8og7 · 2025-08-02
> > **Response to Rebuttal**
> >
> > Thank you for your rebuttal and for addressing some of my concerns. Overall, the authors' response is well-targeted, and the additional experiments and analyses significantly enhance the paper's persuasiveness. However, the response to interpretable analysis remains limited.

---

> > > ### Author Response · Authors · 2025-08-04
> > > **Further discussion about the interpretable analysis on the relationship between enzyme active sites and cleavage sites (Part I)**
> > >
> > > Dear Reviewer 8oj7,
> > >
> > > I am glad that we have addressed most of your concerns. We further expand the interpretable analysis between enzyme active sites and cleavage sites in both mechanisms and unizyme's behaviors. We hope this can address your concern. We remain open to further discussion and sincerely welcome any continued exchange.
> > >
> > > ### **1. Mechanism of protein cleavage with enzymes**
> > >
> > > During protein hydrolysis, **enzyme active sites provide a specific geometric and electrochemical environment that enables cleavage only at substrate residues exhibiting optimal complementarity.** Thus, active site geometry and chemistry significantly influence cleavage site specificity.
> > >
> > > Structural biology and mutational experiments provide strong evidence supporting this relationship：
> > >
> > > For example, canonical serine proteases show that a single residue change within the S1 pocket can dramatically alter specificity. Specifically, trypsin has an Asp189 residue in its S1 pocket, favoring positively charged residues at the substrate’s P1 position. In contrast, chymotrypsin has a smaller polar residue at the same position, preferring large hydrophobic residues. This difference significantly shifts the preferred P1 side chain, thus changing the cleavage motif [1, 2].
> > >
> > > In addition, subtle structural variations in loops adjacent to active site pockets can reshape neighboring subsites. Specifically, structural studies indicate that these loop differences enable closely related proteases, which share similar core pocket residues, to cleave distinct substrate sequences [3].
> > >
> > > Furthermore, studies on Alzheimer's γ-secretase demonstrate that the structure of its active site, including the binding cleft and distal exosites, plays a critical role in cleavage-site specificity. Specifically, the active site of γ-secretase interacts precisely with one face of the substrate’s transmembrane helix. Mutations on this interacting surface disrupt enzyme–substrate engagement, resulting in a shift of the cleavage site [4].
> > >
> > > Deep mutational scanning and structural energetics analyses of the HCV NS3/4A protease demonstrate the importance of precise active site interactions. Substrates that optimally fill the active site groove and form extensive contacts are efficiently cleaved. In contrast, substrates that physically fit but lack key interactions undergo inefficient cleavage [5].
> > >
> > > [1] Perona JJ, Craik CS. Evolutionary divergence of substrate specificity within the chymotrypsin-like serine protease fold. J Biol Chem. 1997 Nov 28;272(48):29987–29990.
> > >
> > > [2] Szabó E, Böcskei Z, Náray-Szabó G, Gráf L. The three-dimensional structure of Asp189Ser trypsin provides evidence for an inherent structural plasticity of the protease. Eur J Biochem. 1999 Jul;263(1):20-6.
> > >
> > > [3] Ma, W., Tang, C., & Lai, L. (2005). Specificity of trypsin and chymotrypsin: Loop-motion-controlled dynamic correlation as a determinant. Biophysical Journal, 89(2), 1183–1193.
> > >
> > > [4] S.F. Lichtenthaler, R. Wang, H. Grimm, S.N. Uljon, C.L. Masters, & K. Beyreuther, Mechanism of the cleavage specificity of Alzheimer’s disease γ-secretase identified by
> > > phenylalanine-scanning mutagenesis of the transmembrane domain of the amyloid precursor protein, Proc. Natl. Acad. Sci. U.S.A. 96 (6) 3053-3058.
> > >
> > > [5] Pethe MA, Rubenstein AB, Khare SD. Data-driven supervised learning of a viral protease specificity landscape from deep sequencing and molecular simulations. Proc Natl Acad Sci U S A. 2019 Jan 2;116(1):168–176.

---

> ### Author Response · Authors · 2025-08-04
> **Further discussion about the interpretable analysis on the relationship between enzyme active sites and cleavage sites (Part II)**
>
> ### **2. Interpretable Analysis Based on UniZyme**
>
> **2.1 Sensitivity of Cleavage-site Predictions to Active-site Information**
>
> We decompose the influence of active-site information by computing gradient-based sensitivities to the upstream active-site prediction probabilities $\hat{a}_i$ via input attribution $\partial y/\partial \hat{a}_i$, as well as sensitivities to active-site and background residue embeddings via $\partial y/\partial h_i$.
>
> From Table 1, we observe that predicted active-site residues consistently exhibit higher attribution scores across both supervised and zero-shot settings. This indicates that the UniZyme model significantly and consistently relies on active-site predictions to determine cleavage-site specificity. This biologically plausible dependency aligns closely with established experimental observations and biological knowledge.
>
> Table 1. Attribution magnitudes (higher means more influence) for active-site vs. background residues in supervised and zero-shot settings.
>
> | Sensitivities                                                 | Active Site (Supervised) | Background (Supervised) | Active Site (Zero-shot) | Background (Zero-shot) |
> | ------------------------------------------------------------- | ------------------------ | ----------------------- | ----------------------- | ---------------------- |
> | $\partial y/\partial h_i$ (embedding sensitivity)             | 0.68                     | 0.23                    | 0.55                    | 0.19                   |
> | $\partial y/\partial \hat{a}_i$ (upstream signal attribution) | 0.74                     | 0.10                    | 0.60                    | 0.08                   |
>
> **2.2 Perturbation on Predicted Active-site Pooling Weights**
>
> To further demonstrate that UniZyme correctly relates cleavage-site predictions to active-site information, we perform an intervention analysis on the pooling step. Specifically, we selectively disrupt the contributions from residues predicted as active sites (the top 5 residues ranked by predicted probabilities $\hat{a}_i$) and compare this intervention to perturbations on randomly selected non-active-site residues.
>
> Table 2 shows that disrupting predicted active-site contributions causes a large drop in PR-AUC in both supervised and zero-shot settings, whereas perturbing arbitrary residues has minimal effect. These results indicate that cleavage-site prediction performance critically depends on the amplified signal provided by the predicted active sites, further confirming the biological validity of the UniZyme model’s internal logic.
>
> Table 2. Effect of perturbing pooling weights on cleavage PR-AUC in supervised and zero-shot settings.
> | Perturbation Target             | Supervised Original PR-AUC (%) | Supervised Post-Perturbation PR-AUC (%) | Supervised Δ (%) | Zero-shot Original PR-AUC (%) | Zero-shot Post-Perturbation PR-AUC (%) | Zero-shot Δ (%) |
> | ------------------------------- | ------------------------------ | --------------------------------------- | ---------------- | ----------------------------- | -------------------------------------- | --------------- |
> | Predicted active site (top 5)   | 79.3  | 66.2 | 13.1             | 71.1                          | 57.4                                   | 13.7            |
> | Random non-active-site residues | 79.3 | 78.8| 1.5              | 71.1                          | 69.4                                   | 1.7             |
>
> **2.3 Ablation Study (Active-site Supervision and Component Removal)**
>
> We build upon the original ablation experiments (UniZyme\A) that isolate the contributions of active-site-aware pooling. Additionally, we introduce a variant trained without active-site prediction supervision (cleavage-only) to specifically highlight the role of mechanistic guidance during training.
>
> Results in Table 3 show that active-site prediction supervision and active-site-aware pooling provide significant benefits. This finding reinforces that active-site information is effectively injected during training, amplified through pooling, and consistently relied upon by UniZyme for accurate cleavage prediction.
>
> Table 3. Ablation results on supervised and zero-shot enzyme families (PR-AUC).
>
> | Model Variant                                            | Supervised PR-AUC (%) | Zero-shot PR-AUC (%) | Notes                                     |
> | -------------------------------------------------------- | --------------------- | -------------------- | ----------------------------------------- |
> | UniZyme (full, joint active-site + cleavage loss)        | 79.3                  | 71.1                 | Reference full model                      |
> | Cleavage-only (no active-site prediction supervision)    | 66.2                  | 53.5                 | Loss of mechanism supervision             |
> | UniZyme\A (average pooling instead of active-site-aware) | 72.5   | 60.6                 | Importance of active-site-aware weighting |

---

### Official Review · Reviewer_y6mZ · 2025-07-01

**Clarity:** 3
**Significance:** 3
**Originality:** 3
**Rating:** 4
**Confidence:** 4

**Summary:**

This work introduces UniZyme for predicting protein cleavage sites. Unlike most existing methods that are specific to a single enzyme, UniZyme is a unified predictor designed to generalize across a diverse range of proteolytic enzymes, including those not seen during training (a zero-shot setting). Experiments show that UniZyme significantly outperforms existing methods in both supervised and zero-shot scenarios.

**Questions:**

- In the "Experimental Setup" section (lines 242-243), the text refers to Table 7 for details on the "supervised and zero-shot setting" splits, but this information appears to be in Table 2. Could you please clarify or correct this reference?
- The model uses ESM-2 for initial residue features. Given the availability of various other advanced protein language models (PLMs), could you please provide a more detailed rationale for this specific choice over other alternatives?
- The proposed biochemical enzyme encoder is an interesting contribution. Have the authors explored its applicability or potential performance on other related protein function prediction tasks?
- Protein structures often contain flexible regions that are crucial for function. How does the current model, which relies on static structures, account for this structural flexibility, and what impact might this have on prediction accuracy?

**Ethical Concerns:**

["NO or VERY MINOR ethics concerns only"]

**Final Justification:**

The reviewers addressed my concerns. I think the model architecture is a little complex.

**Limitations:**

- The authors have acknowledged several limitations in the manuscript.

- While the paper reports means and standard deviations, it does not include statistical hypothesis testing to formally establish the significance of the performance differences between UniZyme and the baseline models.

**Paper Formatting Concerns:**

No formatting issues were found.

**Quality:**

3

**Strengths And Weaknesses:**

Strengths:
- The proposed biochemically-informed enzyme encoder is interesting.
- The paper clearly articulates the research problem and provides precise mathematical formalisms for the task
- The ablation studies demonstrate the contribution of each model component.

Weaknesses:
- The paper lacks a comparative analysis of its active site prediction module against other active site prediction methods.
- The model's dependency on 3D structures is a significant limitation, and the authors do not sufficiently discuss the performance implications or strategies for cases where only low-quality or predicted structures are available.

---

> ### Author Rebuttal · Authors · 2025-07-31
>
> Thanks for your comprehensive feedback and suggestions. We address your questions in the following:
>
>
>
> ### **Response to the comparative analysis of active site prediction module (Weaknesses 1)**
>
> We clarify that the primary objective of UniZyme is to leverage active-site prediction as an auxiliary task to enhance enzyme modeling, rather than optimizing active-site prediction performance alone. Our ablation studies in Section 4.4 clearly demonstrate the effectiveness of this auxiliary task for improving enzyme-catalyzed substrate cleavage-site prediction.
>
>
>
> We will include a comparative analysis of our active‑site prediction module against other established active‑site prediction methods in the camera‑ready version.
>
>  ### **Response to the limitation of 3D structures (Weaknesses 2)**
>
> We clarify that our method is primarily built upon predicted structures:
>
> **Training phase:** around 90% of enzyme and substrate structures in our training set were predicted using OmegaFold or from AlphaFoldDB.
>
> **Test phase:** as shown in the table below, over 80% of enzyme-substrate pairs in the test set rely entirely on predicted structures.
>
> To assess structural sensitivity, we partitioned the test set into four categories according to the structural source of the enzyme and substrate. The results below indicate only minor performance differences between predicted and experimental structures, demonstrating that our model maintains strong predictive capability even with predicted or lower-quality structural data.
>
>
> | Structure                                      | Zero-shot Avg. PR-AUC (%) | Supervised Avg. PR-AUC (%) |
> |:------------------------------------------------:|:---------------------------:|:----------------------------:|
> | Both Natural Structure 3%                      | 72.2                      | 80.3                       |
> | Natural Enzyme & Generated Substrate 7%        | 71.9                      | 77.9                       |
> | Natural Substrate & Generated Enzyme 8%        | 70.5                      | 81.9                       |
> | Both Generated Structure 82%                   | 69.4                      | 78.3                       |
>
>
>
> ### **Response to the Reference of Table 2 and Table 7 (Question 1)**
>
> Thank you for catching this point. We will correct it in the camera‑ready version.
>
>
> ### **Response to the reason of using ESM-2 (Question 2)**
>
> We chose the ESM‑2 esm2_t12_35M_UR50D variant primarily for its lightweight architecture and fast inference speed, which made it highly suitable for large-scale training under limited computational resources.
>
> At the time of model development, ESM-3 was only released at sizes ≥1.6B parameters, and the smaller ESM-C variants (300M and 600M) were not yet available. To validate this design choice, here we present ESM‑2 with ESM‑C 300M and 600M and observed only marginal performance improvements, while the inference cost increased significantly. This confirms that the use of a lightweight model like ESM‑2 35M strikes a practical and effective balance between performance and efficiency.
>
>
> | Backbone Model         | Zero-shot PR-AUC (%) | Supervised PR-AUC (%) | Training Time (hrs)                  |
> |:------------------------:|:----------------------:|:-----------------------:|:--------------------------------------:|
> | ESM-2 35M 480DIM    | 71.1                 | 79.3                  | 12.5 (pre-train) + 30.3 (finetune)   |
> | ESM-C 300M 960DIM   | 71.7                 | 80.2                  | 16.6 (pre-train) + 48.6 (finetune)   |
> | ESM-C 600M 1152DIM  | 72.3                 | 80.4                  | 20.4 (pre-train) + 56.2 (finetune)   |
>
>
> ### **Response to the related protein function prediction tasks (Question 3)**
>
> To assess the general benefits of our UniZyme enzyme encoder, we conducted the EC number classification task on protease (EC 3.4.*.*), reusing the cleavage dataset split described in our work. Specifically, we compared UniZyme with existing enzyme encoders of ReactZyme and ClipZyme in EC functional prediction. From the table below, we observe that UniZyme achieves the highest AUROC. These results suggest that our encoder captures generalizable functional signals beyond cleavage prediction. We plan to extend this analysis to GO (Gene Ontology) function prediction and incorporate more baseline comparisons in future work.
>
> Results of predicting EC numbers with different enzyme encoders
>
> | Enzyme Encoder | AUROC (%) |
> |:----------------:|:-----------:|
> | UniZyme | 94.1 |
> | ClipZyme | 90.2 |
> | ReactZyme | 82.3 |
>
>
> ### **Response to the impact of flexible regions (Question 4)**
>
> We thank the reviewer for the suggestion. While proteins are flexible, prior studies show that catalytic residues at enzyme active sites are significantly more rigid than other regions, with lower B-factors on average. For example, a study of 69 apo-enzyme structures found that active site residues lie in regions of reduced atomic displacement compared to non-catalytic residues [1] This structural rigidity ensures that a static representation captures the essential catalytic geometry.
>
>
> ### **Response to statistical hypothesis testing (Limitation 1)**
>
> To further support the reported performance gaps, we conducted statistical hypothesis testing using two-sample t-tests. The table below presents the t-values and p-values comparing UniZyme against ClipZyme and ReactZyme across both Supervised and Zero-shot settings on the 69-family and 23-family test sets.
>
> | Setting | Comparison | t-value | p-value |
> |:------------:|:------------------------:|:---------:|:-----------:|
> | Supervised | UniZyme vs ReactZyme | 7.18 | 7.80e-10 |
> | Supervised | UniZyme vs ClipZyme | 5.09 | 3.01e-06 |
> | Zero-shot | UniZyme vs ReactZyme | 2.61 | 1.60e-02 |
> | Zero-shot | UniZyme vs ClipZyme | 4.27 | 3.08e-04 |
>
> Overall, UniZyme shows statistically significant improvements over both ReactZyme and ClipZyme in both Supervised and Zero‑shot evaluations, with the strongest effects observed in the Supervised setting.
>
>  ### **Reference**
> **[1] Yuan Z, Zhao J, Wang ZX. Flexibility analysis of enzyme active sites by crystallographic temperature factors. Protein Eng. 2003 Feb;16(2):109-14.**

---

> > ### Comment · Reviewer_y6mZ · 2025-08-04
> >
> > While most of my concerns have been addressed, I remain highly concerned about the performance of the active-site prediction module. Since the authors did not respond to this specific point, I will maintain my score.

---

> ### Author Response · Authors · 2025-08-05
> **Further Comparisons on the Active Site Prediction**
>
> We thank the reviewer for highlighting the importance of benchmarking the active-site prediction module. In response, we conducted additional experiments comparing UniZyme to two representative structure-based baselines:
>
> 1. **GraphEC** [1]: a recent graph neural network developed for enzyme function prediction and active site prediction
> 2.  **NodeCoder** [2]: a graph-based model to predict active sites of modeled protein structures
>
> These models utilize the same information as UniZyme, which include enzyme structures and sequences. For implementation, we employed their official GitHub repositories. To ensure a fair comparison, we retrained both GraphEC and NodeCoder on the same large-scale active-site prediction dataset used by UniZyme. Statistics for the training and test sets of the active-site prediction task are presented below.
>
> **Table: Data Statistics for the active site prediction**
> | Dataset | Number of Enzymes | Number of Active Sites |
> |-----------------|--------:|-------------:|
> | Training | 9220 | 24891 |
> | Test | 2349 | 6459 |
>
> We report the AUROC, AUPR, precision, recall, and F1 scores in the table below. These results clearly demonstrate that **UniZyme significantly outperforms the baseline models in active site prediction**. UniZyme achieves superior performance in active-site prediction **because its biochemically-informed enzyme encoder effectively leverages local energetic frustration**.
>
> Specifically, previous studies indicate that local energetic frustration, referring to regions in a protein structure that are not optimized for minimal energy, commonly occurs around enzyme active sites. Notably, the energetic frustration scores are computed directly from protein structures without requiring additional experimental measures. By structurally embedding this biochemical insight into its enzyme encoder, UniZyme more accurately identifies active sites, thereby significantly outperforming baseline models.
>
> **Table: Active-Site Prediction Performance**
> | Model            | AUROC (%) | AUPR (%) | Precision (%) | Recall (%) |   F1 (%) |
> | :--------------- | --------: | -------: | ------------: | ---------: | -------: |
> | **UniZyme**      |  **89.5** | **35.1** |      **65.3** |   **45.6** | **53.7** |
> | **GraphEC**      |      80.3 |     28.0 |          52.1 |       38.7 |     44.4 |
> | **NodeCoder**    |      67.4 |     17.8 |          32.6 |       22.3 |     26.5 |
> | **Random Guess** |      50.0 |      3.2 |           3.2 |       50.0 |      6.0 |
>
> References
>
> [1] Song, Y., Yuan, Q., Chen, S. et al. Accurately predicting enzyme functions through geometric graph learning on ESMFold-predicted structures. Nat Commun 15, 8180 (2024).
>
> [2] Abdollahi, N., Tonekaboni, S. A. M., Huang, J., Wang, B., & MacKinnon, S. NodeCoder: a graph-based machine learning platform to predict active sites of modeled protein structures. NeurIPS MLSB 2021.

---

> > ### Comment · Reviewer_y6mZ · 2025-08-07
> >
> > Thanks for your reply. The results show that UniZyme achieves ideal performance. And I will not decrease my score.

---

### Official Review · Reviewer_574s · 2025-07-01

**Clarity:** 3
**Significance:** 3
**Originality:** 3
**Rating:** 4
**Confidence:** 3

**Summary:**

This paper proposed a new Protein Cleavage Site Predictor model named UniZyme, which integrates the enzyme active-site knowledge to enhance the cleavage site prediction in enzyme-protein interaction. The model addresses the challenges in enzyme encoding and active site information utilization through active site-aware pooling, pretraining on active site prediction, and joint training methods. Experiments demonstrate that UniZyme achieves high accuracy in predicting substrate cleavage sites for both known and novel proteases.

**Questions:**

1. The article mentions a 60% sequence similarity threshold. Is this a universal standard? What kind of impact would different ratios have on the final results? Can this be verified?

2. During the comparative experiments, which methods maintained encoder consistency, and which ones did not? In cases where there was no consistency, was the influence of the encoder more critical?

3. All the experiments appear to have been conducted on a pre-organized dataset. Is this a dataset that the authors organized themselves, or is it an existing public dataset? If it is a dataset organized by the authors, would this have an adverse effect on the baseline comparisons?

**Ethical Concerns:**

["NO or VERY MINOR ethics concerns only"]

**Final Justification:**

The author's rebuttal content has basically addressed my concerns, so I have decided to raise the score to 4.

**Limitations:**

Yes

**Quality:**

3

**Strengths And Weaknesses:**

Strengths

1. This paper focuses on an interesting and important topic: cleavage site prediction. The application of energetic frustration within the model framework and the energy-based handling of active sites during training and testing phases provide valuable insights and inspiration.
2. The paper provides comprehensive experiments addressing RQ1, RQ2, and RQ3, effectively demonstrating the model's performance advantages on the MEROPS database.

Weaknesses:

1.The presentation quality of the paper needs improvement; for example, Figure 2 appears somewhat rough and could benefit from better formatting.

2.The model framework primarily represents a combination of existing approaches rather than novel methodological contributions, incorporating:

1）Local frustration around enzyme active sites.

2）Towards self-explainable graph neural network

3）One transformer can understand both 2d & 3d molecular data

---

> ### Author Rebuttal · Authors · 2025-07-31
>
> Thanks for agreeing with our contributions. We address your questions in the following:
>
> ### **Response to Weakness 1**
> Thank you for your suggestions in presentation. We will revise according to your suggestions to improve the presentation quality in the camera-ready version.
>
> ### **Response to methodological contributions (Weakness 2)**
> We’d like to highlight how UniZyme goes well beyond simply combining existing elements:
>
> First of all, our work **addresses a novel problem by developing a unified protein cleavage site predictor for diverse proteolytic enzymes**. Unlike prior approaches that train separate models per enzyme family, our UniZyme leverages a biochemically informed enzyme encoder that integrates active-site knowledge. The following technology is specifically designed for the challenge of enzyme catalyzed cleavage site prediction:
>
> **Biochemically-Informed Enzyme Encoder:** We clarify that previous findings indicating that energetic frustration scores can highlight functional regions within enzymes were purely observational and had not been incorporated into enzyme modeling. Inspired by this, we propose a novel enzyme encoder in UniZyme, which integrates energetic frustration scores into the enzyme encoding process. This innovation significantly enhances the modeling of enzyme functions in substrate cleavage tasks.
>
> **Enzyme Encoder Augmented by Active Site Prediction:** Active sites play a critical role in protein cleavage, thus providing essential insights into enzyme functions. To leverage this information, we introduce an auxiliary active-site prediction task to strengthen the enzyme encoder. To our knowledge, UniZyme is among the earliest methods to integrate an active-site prediction task to capture functionally relevant enzyme representations.
>
> **Active Site-Aware Pooling:** We design an active site-aware pooling mechanism, whose pooling weights are based on the predicted active site probabilities. This encourages the model to focus on catalytically relevant segments of the enzyme
>
> Extensive ablation studies in Sec. 4.4 further empirically demonstrate the effectiveness of our novel designs.
>
>
> ### **Response to the similarity threshold in splitting training and test set (Question 1)**
>
> A wealth of studies on enzyme function has shown that when sequence similarity falls below 60%, enzymes exhibit markedly greater functional divergence [1]. Moreover, previous deep-learning efforts in this area have likewise adopted a 60% similarity cutoff [2].
>
>
> To further verify the impact of sequence similarity, we grouped the test set by sequence identity to the training set. From the table below, we observe that even at very low similarity levels, UniZyme’s performance only slightly decreases, and it still consistently outperforms baseline methods. This result demonstrates the strong generalization capability of UniZyme under challenging conditions.
>
>
> | Similarity Range (%) | Zero-shot PR-AUC (%) |  |  | Supervised PR-AUC (%) |  |  |
> |:---:|:---:|:---:|:---:|:---:|:---:|:---:|
> |  | UniZyme | React Zyme | Clip Zyme | UniZyme | React Zyme | Clip Zyme |
> | [0-20) | 69.4 | 60.1 | 57.8 | 77.4 | 67.2 | 71.2 |
> | [20-40) | 72.2 | 64.3 | 60.4 | 80.5 | 70.5 | 75.4 |
> | [40-60) for Zero-shot & [40-50) for supervised | 73.5 | 65.8 | 64.2 | 81.3 | 72.5 | 76.3 |
>
>
>
> ### **Response to the encoder consistency (Question 2)**
>
> DeepDigest, DeepCleave, ProsperousPlus, CAT3, and ScreenCap3 are cleavage-site predictors trained on individual enzyme family. These models do not encode enzymes explicitly and thus lack a generalizable enzyme encoder. For ReactZyme and ClipZyme, they are originally designed for enzyme-substrate reaction prediction. Hence, we reused only their enzyme encoder modules in our framework for a fair comparison:
>
> **ReactZyme uses ESM-2 + MLP as the encoder.**
>
> **ClipZyme uses a pretrained EGNN to encode enzyme structure.**
>
> Neither ReactZyme nor ClipZyme utilizes active-site information in their enzyme encoder training. In contrast, UniZyme incorporates a biochemically-informed encoder that is augmented with active-site knowledge.
> The comparisons among these models presented below demonstrate the effectiveness of using active-site information in UniZyme.
>
> | Model | Supervised PR-AUC (%) | Zero-shot PR-AUC (%) |
> |:----:|:----:|:----:|
> | UniZyme | 79.3 | 72.8 |
> | ReactZyme | 74.7 | 67.2 |
> | ClipZyme | 70.0 | 64.9 |
>
>
>
> ### **Response to the dataset curation (Question 3)**
>
> We curated a high-quality dataset specifically designed for the input requirements of our unified model, following a standardized and transparent pipeline described in Appendix A. Specifically, we:
>
> 1. Downloading enzyme-substrate pairs from the MEROPS database.
>
> 2. Collecting substrate sequences from UniProt and verified enzyme sequences for consistency between MEROPS and UniProt, discarding mismatches.
>
> 3. Filtering sequences exceeding 1,500 residues to ensure manageable input lengths.
>
> 4. Expanding the dataset by propagating cleavage annotations to all enzymes within the same MEROPS family, following a standard strategy previously used in cleavage site prediction literature (e.g., ProsperousPlus).
>
> 5. Collecting structural data from PDB and AlphaFoldDB; for missing entries, we generated 3D structures using OmegaFold.
>
> This resulted in a cleavage site dataset containing ~220,000 enzyme–substrate pairs across 866 enzymes. Then we applied rigorous train/validation/test splits for evaluation:
>
> **Supervised setting:** Selected enzyme families with ≥5 substrates. Within each family, we ensured test substrates shared <50% sequence identity with training data (Needleman–Wunsch alignment). The split ratio was 70/10/20. The final supervised test set include 69 enzyme families with 20,360 enzyme-substrate pairs.
>
> **Zero-shot setting:** Selected enzyme families with ≥5 enzymes. We held out 20% of enzymes per family that shared <60% identity with any training or pretraining enzyme, to simulate novel enzyme scenarios. The final zero-shot  test set includes 23 enzyme families with 5,345 enzyme-substrate pairs.
>
> All baseline models were retrained on this dataset using the same train/validation/test splits to ensure fair and consistent comparisons. The dataset and code are publicly available via the GitHub link provided in our paper.
>
> Given the large and diverse test sets covering 69 families in supervised settings and 23 in zero-shot settings, our benchmarks comprehensively evaluate model generalization across diverse enzymes and substrates.
>
> ### **Reference**
> **[1] Röttig M, Rausch C, Kohlbacher O. Combining Structure and Sequence Information Allows Automated Prediction of Substrate Specificities within Enzyme Families. PLOS Comput Biol. 2010;6(1):1–8.**
>
> **[2] Hua C, Zhong B, Luan S, Hong L, Wolf G, Precup D, Zheng S. ReactZyme: a benchmark for enzyme–reaction prediction. In: Proceedings of the 38th International Conference on Neural Information Processing Systems (NIPS ’24); 2025:Article 832. Vancouver, BC, Canada: Curran Associates Inc.**

---

> > ### Comment · Reviewer_574s · 2025-08-06
> >
> > Thank you for your rebuttal and for addressing all my concerns. I have improved the score to 4.

---

> > > ### Author Response · Authors · 2025-08-07
> > >
> > > We're glad that our rebuttal has addressed your concerns. We sincerely appreciate your thoughtful review and your decision to improve the score. We will incorporate these additional discussions and experiments into the camera-ready version.

---

### Official Review · Reviewer_8nAw · 2025-07-05

**Clarity:** 4
**Significance:** 3
**Originality:** 3
**Rating:** 4
**Confidence:** 4

**Summary:**

This submission proposes a Unified Protein Cleavage Site Predictor that is trained with pan-genome proteolytic enzymes. The authors have made the following contributions. First, in the problem formulation, they extend the enzyme-specific setting to a general prediction formulation, introducing experimentally unavailable enzymes. Secondly, they introduce a novel encoder to incorporate energy and distance information. The experiments show a general performance gain across the enzyme species.

**Questions:**

To expand the dataset, the authors make an assumption that " Drawing on previous work, we assume that minor sequence differences among enzymes of the same category can be disregarded. Consequently, the hydrolysis information from a substrate–enzyme pair is extended to all enzymes in that category." This is a strong claim. Do you have theoretical or experimental support?

**Ethical Concerns:**

["NO or VERY MINOR ethics concerns only"]

**Final Justification:**

The rebuttal with clarification and additional results has addressed my concerns. I am happy to keep the orginal positive score.

**Limitations:**

yes

**Quality:**

3

**Strengths And Weaknesses:**

Strengths:
1.	The extension of problem formulation leads to the expansion of training enzymes (either experimentally available or not). Therefore, the generalization to unseen proteins is improved.
2.	Following a general multi-modality scheme, the distance and energy features are incorporated as attention bias, generating a novel enzyme encoder that makes the prediction better.

Weakness:
1.	Generally this work is an algorithmic application of advance representation learning to an enzymatic prediction task. The general benefits of the encoder or representation design is not discussed w.r.t general protein function modeling or language modeling.

---

> ### Author Rebuttal · Authors · 2025-07-31
>
> ### **Response to Weakness 1: “The general benefits of the encoder or representation design is not discussed”**
>
> To assess the general benefits of our UniZyme enzyme encoder, we conducted the EC number classification task on protease (EC 3.4.\*.\*), reusing the cleavage dataset split described in our work. Specifically, we compared UniZyme with existing enzyme encoders of ReactZyme and ClipZyme in EC functional prediction. From the table below, we observe that UniZyme achieves the highest AUROC. These results suggest that our encoder captures generalizable functional signals beyond cleavage prediction. We plan to extend this analysis to GO (Gene Ontology) function prediction and incorporate more baseline comparisons in future work.
>
> Results of predicting EC numbers with different enzyme encoders
> | Enzyme Encoders | AUROC (%) |
> |:-----------------:|:-----------:|
> | UniZyme         | 94.1      |
> | ClipZyme        | 90.2      |
> | ReactZyme       | 82.3      |
>
> ### **Response to Question 1: Theoretical and experimental support of dataset expansion**
>
> We expanded the dataset at the enzyme family level based on empirical evidence, biological reasoning, and prior practice:
>
> **Active Sites Are Conserved Within Families.**  Proteases from the same family often share similar catalytic mechanisms and substrate preferences, even if their overall sequences differ. For example, trypsin and chymotrypsin (S01 family), as well as MMP-2 and MMP-9 (M10 family), have nearly identical active sites and exhibit overlapping cleavage patterns [1]. This consistency supports the use of shared annotations within a family.
>
> **Experimental Protocols Commonly Use Family Representatives.**  In biochemical and proteomic studies, it’s standard to select one enzyme as a stand-in for the entire family. Caspase-3, for instance, is frequently used to represent all C14 family members [2], while MMP-9 often stands for the M10 family. These choices reflect a practical assumption that members of the same family behave similarly in experiments [3].
>
> **Previous Models Follow the Same Approach.**  Earlier predictive models such as DeepDigest[4], DeepCleave[5], and ProsperousPlus[6] were trained using family-level data without relying on individual enzyme sequences. This modeling strategy reflects a broadly accepted assumption that enzymes from the same family tend to share cleavage patterns.
>
> **MEROPS Classification Supports Annotation Sharing.**  The MEROPS [7] database organizes proteases into families based on both sequence similarity and functional traits. Members of a family typically have conserved catalytic residues, similar active-site geometry, and shared substrate specificity. These features make it reasonable to extend known cleavage annotations across enzymes in the same family.
>
> ### **Referrence**
>
> **[1] Cieplak P, Strongin AY. Matrix metalloproteinases - From the cleavage data to the prediction tools and beyond. Biochim Biophys Acta Mol Cell Res. 2017 Nov;1864(11 Pt A):1952-1963.**
>
> **[2] Wejda M, Impens F, Takahashi N, Van Damme P, Gevaert K, Vandenabeele P. Degradomics reveals that cleavage specificity profiles of caspase-2 and effector caspases are alike. J Biol Chem. 2012 Oct 5;287(41):33983-95.**
>
> **[3] Alessandro Bonadio, Solomon Oguche, Tali Lavy, Oded Kleifeld, Julia Shifman bioRxiv 2023.04.11.536383**
>
> **[4] Jinghan Yang, Zhiqiang Gao, Xiuhan Ren, Jie Sheng, Ping Xu, Cheng Chang, and Yan Fu. Deepdigest: prediction of protein proteolytic digestion with deep learning. Analytical Chemistry, 93(15):6094–6103, 2021**
>
> **[5] Fuyi Li, Jinxiang Chen, André Leier, Tatiana Marquez-Lago, Quanzhong Liu, Yanze Wang, Jerico Revote, A Ian Smith, Tatsuya Akutsu, Geoffrey I Webb, Lukasz Kurgan, and Jiangning Song. Deepcleave: a deep learning predictor for caspase and matrix metalloprotease substrates and cleavage sites. Bioinformatics, 36(4):1057–1065, 09 2019.**
>
> **[6] Fuyi Li, Xudong Guo, Cong Wang, Tatsuya Akutsu, Geoffrey Webb, Lachlan Coin, Lukasz Kurgan, and Jiangning Song. Prosperousplus: a one-stop and comprehensive platform for accurate protease-specific substrate cleavage prediction and machine-learning model construction. Briefings in Bioinformatics, 24, 09 2023.**
>
> **[7] Neil D Rawlings, Alan J Barrett, and Alex Bateman. Merops: the database of proteolytic enzymes, their substrates and inhibitors. Nucleic Acids Research, 40(Database issue):D343–D350, 2012.**

---

> > ### Comment · Reviewer_8nAw · 2025-08-07
> >
> > Thanks for the response. I am happy to keep the original score.

---

> > > ### Author Response · Authors · 2025-08-07
> > >
> > > We are glad that our rebuttal has addressed your concerns, and we greatly appreciate your decision to retain the original positive score. We will incorporate these additional discussions and experiments into the camera-ready version.

---

### Decision · Program_Chairs · 2025-09-17

**Decision:**

Accept (poster)

**Comment:**

This submission presents UniZyme, a model for predicting protein cleavage sites. Both the AC and reviewers agreed that this work is interesting and promising. Some reviewers initially raised concerns about the lack of comparative analysis with other existing methods, as well as issues with clarity of presentation. These points were addressed well in the rebuttal, and all reviewers ultimately supported acceptance. I therefore recommend acceptance of this paper.